METHODS AND RESOURCES

# *morphoHeart*: A quantitative tool for integrated 3D morphometric analyses of heart and ECM during embryonic development

**Juliana Sánchez-Posada**[1]*, **Christopher J. Derrick**[2], **Emily S. Noël**[1]*

**1** School of Biosciences and Bateson Centre, University of Sheffield, Western Bank, Sheffield, United Kingdom, **2** Biosciences Institute, Faculty of Biomedical Sciences, Newcastle University, International Centre for Life, Central Parkway, Newcastle upon Tyne, United Kingdom

* jsanchezposadam@gmail.com (JS-P); e.s.noel@sheffield.ac.uk (ESN)

## Abstract

Heart development involves the complex structural remodelling of a linear heart tube into an asymmetrically looped and ballooned organ. Previous studies have associated regional expansion of extracellular matrix (ECM) space with tissue morphogenesis during development. We have developed *morphoHeart*, a 3D tissue segmentation and morphometry software with a user-friendly graphical interface (GUI) that delivers the first integrated 3D visualisation and multiparametric analysis of both heart and ECM morphology in live embryos. *morphoHeart* reveals that the ECM undergoes regional dynamic expansion and reduction during cardiac development, concomitant with chamber-specific morphological maturation. We use *morphoHeart* to demonstrate that regionalised ECM expansion driven by the ECM crosslinker Hapln1a promotes atrial lumen expansion during heart development. Finally, *morphoHeart*'s GUI expands its use beyond that of cardiac tissue, allowing its segmentation and morphometric analysis tools to be applied to *z*-stack images of any fluorescently labelled tissue.

**Data Availability Statement:** All relevant data are within the paper and its Supporting Information files. All numerical data can be found in File S1

## Introduction

Tissue morphogenesis in development requires the elaboration of simple structures into complex shapes. This includes common processes such as epithelial folding and tubular morphogenesis, requiring coordinated growth and shaping of multiple tissue layers or cell types. The developing heart is an excellent example of such a morphogenetic transformation. The embryonic heart initially forms a linear tube that comprises an outer myocardial tube and inner endothelial lining (the endocardium). This linear tube undergoes a complex morphogenesis that includes bending and looping of the tube to align the segments of the heart, and regional ballooning of the tube to start forming the chambers [1]. This morphogenesis is vital for establishing the blueprint of the heart and is followed by substantive organ growth and formation of

Data. The current version of morphoHeart software can be downloaded from Github at https://github.com/jsanchez679/morphoHeart. The archived version of morphoHeart at acceptance can be downloaded from Zenodo at https://zenodo.org/records/14480354.

**Funding:** E.N was supported by a British Heart Foundation Fellowship (www.bhf.org.uk) award FS/16/37/32347. E.N and J.S-P were supported by BBSRC (www.https://www.ukri.org/councils/bbsrc) Standard Grant BB/W004305/1. Lightsheet imaging was performed at the Wolfson Light Microscopy Facility using Zeiss Z1 lightsheet microscopes funded through a BBSRC (www.https://www.ukri.org/councils/bbsrc) ALERT14 award BB/M012522/1 and a BHF (www.bhf.org.uk) Infrastructure Grant IG/15/1/31328. The funders had no role in study design, data collection and analysis, decision to publish, or preparation of the manuscript.

**Competing interests:** The authors have declared that no competing interests exist.

**Abbreviations:** dpf, days post fertilisation; ECM, extracellular matrix; GUI, graphical user interface; HA, hyaluronic acid; hpf, hours post fertilisation; IND, internuclear distance; MIP, Maximum Intensity Projection; PG, proteoglycan; ROI, region of interest; SHF, second heart field.

structures such as valves and trabeculae to support function. Therefore, defects in early heart morphogenesis can have profound impacts on later heart structure and function [2].

The extracellular matrix (ECM) is a crucial signalling centre in tissue development including the heart [3–5] where it provides biochemical and biomechanical cues to cardiomyocytes and endocardial cells. During early heart morphogenesis, the cardiac ECM is rich in the glycosaminoglycan hyaluronic acid (HA) and the proteoglycan (PG) versican, both of which play conserved roles in heart morphogenesis [6–10]. Both HA and versican can sequester water [11,12], allowing them to swell the ECM, increasing volume and hydrostatic pressure [13]. Hydrostatic pressure is increasingly recognised as a driver of tissue morphogenesis [14], and HA-mediated expansion of ECM volume is important in several developmental contexts, including epithelial projection formation in the ear [15,16], initiation of atrioventricular valve formation [17], and driving of midgut rotation during gut development [18]. Furthermore, our previous work indicated that the cardiac ECM is asymmetrically expanded prior to heart tube morphogenesis through regionalised expression of the HA and proteoglycan cross-linking protein Hapln1a and that this asymmetric ECM expansion is required for atrial morphogenesis [7]. Thus, analysing ECM-space in conjunction with detailed morphometric descriptors of the adjacent tissue is vital to gain a better understanding of these matrix-tissue relationships during development.

Tools to analyse early heart morphogenesis in detail are limited. Recent studies in mouse have performed detailed quantitative characterisation of fixed samples [19–22]. However, fixed tissue analyses can have limitations, for example, collapse or shrinkage of tissue due to fixation, and alterations to the ECM (by modifying hydration or crosslinking), which may hamper efforts to understand volumetric ECM dynamics in the context of cardiac morphogenesis. Where possible, live analyses would address these issues, but imaging embryos that normally develop *in utero*, such as mouse, is challenging, and has been limited to stage-restricted analysis of embryos in live explant culture [23–25]. Zebrafish represent an excellent model in which to analyse early stages of heart development live: the embryos are transparent, develop externally, and early morphogenesis of the heart tube together with genetic pathways underlying development and disease are well-conserved [26]. Limited morphometric segmentation of the zebrafish heart has been described [27] through manual segmentation of a single tissue layer with a limited number of defined parameters for analysis. While more complex morphometric tissue analyses are becoming more widely adopted, these often use bespoke code, or software that are not able to handle the complexity of the heart and are not able to extract information about the acellular space between tissue layers, such as volumetric reconstructions of the cardiac ECM.

We have taken advantage of the zebrafish model for analysing heart morphogenesis and used it to develop a new open-source image analysis software *morphoHeart*, which allows multiparametric morphometric analysis of the developing heart in live embryos, including segmentation of the cardiac ECM. The design of *morphoHeart's* graphical user interface (GUI) expands its use beyond that of cardiac tissue, facilitating analysis of multiple tissue layers, including label-free negative-space segmentation of tissue or fluid within layers and division of tissue into segments or regions of interest for more granular analysis. Here, we use *morphoHeart* to reveal new insights into early cardiac morphogenesis in the developing zebrafish.

## Design

Developing organs such as the heart are complex tissues, and their detailed morphology can be challenging to interpret using 2D images. Quantitative analyses of cardiac morphology are still limited, and label-free visualisation of ECM volume is not currently possible. Understanding

the embryonic origins of congenital malformations requires refined quantitative analyses of heart morphology. These are generally restricted to fixed tissue, and currently no software can identify and analyse all layers that contribute to the heart.

Therefore, the core design goal for *morphoHeart* was to generate a visualisation and analysis software able to generate 3D reconstructions of the myocardium, endocardium, and cardiac ECM of developing hearts from fluorescently labelled images, facilitating multiparametric morphometric analysis of the tissue layers and heart morphology throughout development. This also includes visualisation and quantitative analysis not only of geometrical and volumetric parameters, but also tissue thickness, cell size, and tissue expansion. Finally, we also wished to visualise conserved morphological features between biological samples, which can be hampered by heterogeneous morphology between individuals, and thus developed a method to standardise and overlay replicates to visualise conserved biological features.

These design goals required the incorporation of multiple methodologies, with features that can be user-configured to make the software adaptable to different needs. For example, the implementation of negative-space segmentation of the ECM required a contour-based approach to create a volumetric reconstruction from the myocardial and endocardial contours. Analysis of tissue deformation required definition of a line running through the centre of a specific 3D tubular surface, and thus we incorporated centreline-finding methodology typically used to describe vasculature. Multi-sample analyses required integrated use of centrelines, user-defined cutting planes, and automated re-slicing to generate quantitative representations of individual and average morphological heart parameters.

*morphoHeart* was designed to be user-friendly, independent of programming experience. While it was originally developed for analysis of cardiac tissue, its GUI is designed to be appropriate for analysis of any type of fluorescently labelled image sample in which tissues can be segmented from contours—including analysis of single or multiple layers, extraction of negative-space volumes, and multi-segment and region analyses.

## Results

### Image acquisition, preprocessing, and morphoHeart reconstructions

To segment the 3 layers of the heart (myocardium, endocardium, and ECM), *Tg(myl7:life-ActGFP); Tg(fli1a:AC-TagRFP)* double transgenic zebrafish embryos were imaged, in which the myocardium and endocardium are labelled. The heartbeat was temporarily arrested, and *z*-stacks encompassing the heart were acquired (S1A–S1C Fig). To remove noise artefacts, accentuate details, and enhance tissue borders, *z*-stacks are then processed, filtered, cropped (S1D–S1F Fig), and imported into *morphoHeart* for tissue layer segmentation (Fig 1A). To extract the myocardium and endocardium, individual slices making up each channel go through a process of automatic contour detection and semi-automatic selection, extracting the contours that delineate the tissue layer (Fig 1B and 1C). Selected contours are classified either as internal or external, depending on whether they outline the lumen or the external borders of the tissue, respectively. Classified internal and external contours are used to create filled binary masks, one containing just the filled lumen of the tissue of interest (henceforth "filled internal contour mask," Fig 1F and 1G), and another containing both the tissue layer, and the filled lumen (henceforth "filled external contour mask," Fig 1D and 1E). For each channel, external and internal contour masks are combined, applying an exclusive disjunction logical operator (XOR) to obtain a final mask comprising just the tissue layer itself (Fig 1H and 1I). Together, these masks constitute the contour library from which all *morphoHeart* operations can subsequently be performed (Fig 1J). The resulting binary masks can be used to create

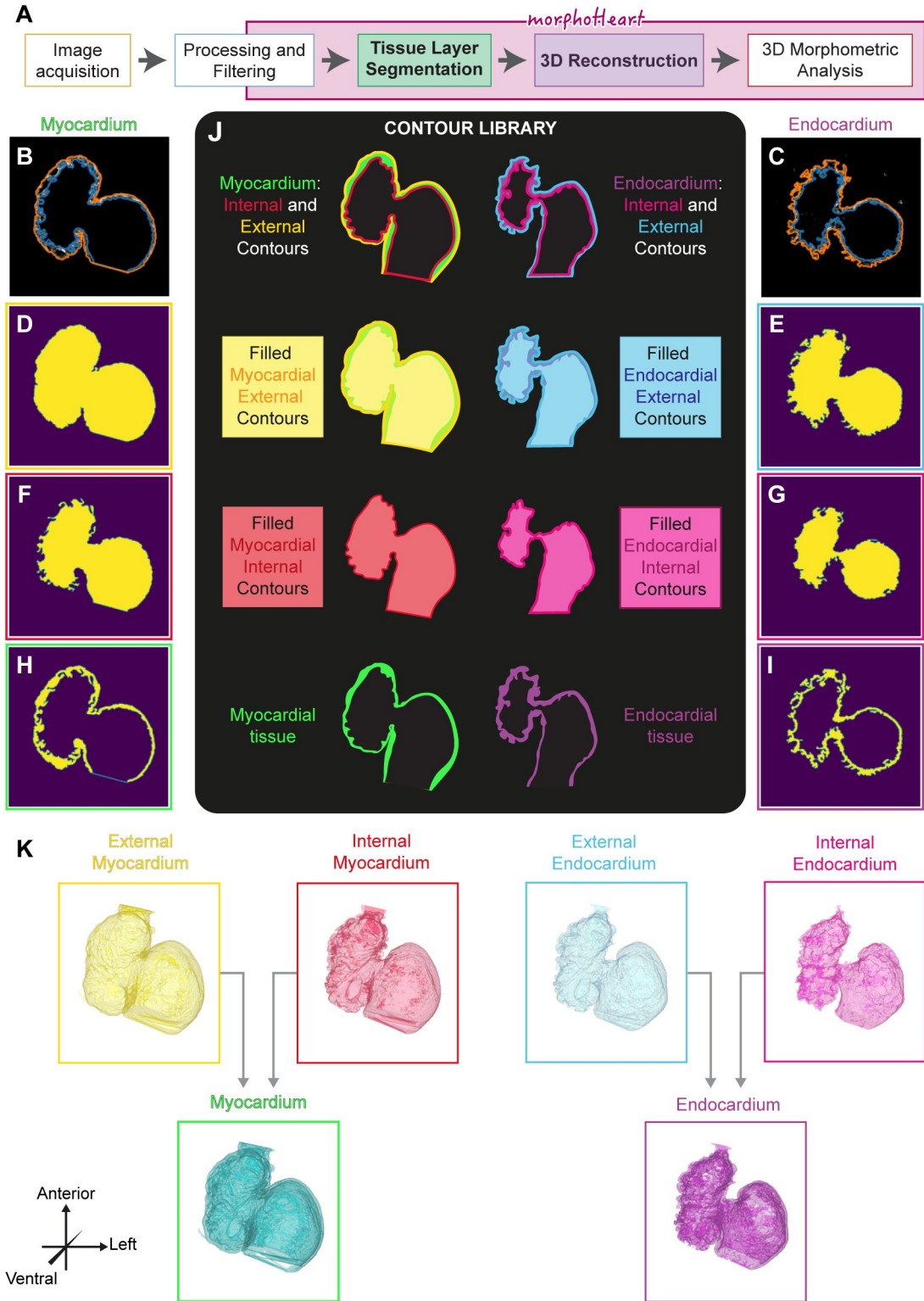

**Fig 1.** ***morphoHeart*** **image segmentation and volumetric tissue reconstructions.** (A) Schematic overview of the *morphoHeart* image processing and segmentation pipeline. (B–I) Generation of tissue contour libraries. Identification of outer (orange) and inner (blue) contours of the myocardium and endocardium in single z-slices (B, C). For each slice masks are generated representing filled outer (D, E) and inner (F, G) tissue contours. The exclusive disjunction operation of inner and outer contours per tissue results in a tissue mask for each slice (H, I). (J) Contour library generated by *morphoHeart*. (K) 3D mesh

reconstructions of external (yellow) and internal (orange) myocardium, external (blue) and internal (pink) endocardium, and myocardial (green) and endocardial (magenta) tissue layers.

volumetric 3D reconstructions (hereafter meshes) of each tissue layer of the heart (Fig 1K), facilitating the extraction of 3D morphological readouts to characterise heart morphogenesis.

## The zebrafish heart undergoes periods of growth and compaction during early morphogenesis

To characterise the morphological changes the heart undergoes throughout looping and ballooning (Fig 2B–2D), we imaged *Tg(myl7:lifeActGFP); Tg(fli1a:AC-TagRFP)* double transgenic zebrafish embryos at early looping (34 hours post fertilisation (hpf) to 36 hpf), after initial looping (48 hpf and 50 hpf), prior to onset of trabeculation (58 hpf to 60 hpf) and during early maturation of the chambers (72 hpf to 74 hpf), generating volumetric meshes of both the myocardium and endocardium (Fig 2E–2H). Various methodologies have been previously described to quantify the extent of looping morphogenesis in zebrafish, relying on 2D image analysis [7,27–29]. Since an early primitive state of the heart is a linear tube, and looping is a morphological deviation from a linear state, we used the linear distance between poles and the length of the heart's centreline to calculate a looping ratio, similar to our previous approach [7], but taking into account the 3D organisation of the tissue. A heart "centreline" (i.e., the 3D curve within the lumen of the heart, whose minimal distance in 3D from the tissue wall is maximal) (Figs 2I and S2A–S2D) was defined through calculation of a Voronoi diagram of the internal surface of the myocardium (S2B Fig, see http://www.vmtk.org/ for more details) and connected to the centre of the venous and arterial poles (which also serve as anchor points to calculate the direct linear distance between the poles). As looping proceeds, the linear distance between the poles reduces (Fig 2J) and between 34 to 50 hpf the length of the centreline (or looped heart length) increases (Fig 2K), together resulting in an increase in the looping ratio (Fig 2L). Interestingly, while the linear distance reduces between 48 and 74 hpf as the heart continues to compress and shorten the distance between poles, the centreline length also decreases, resulting in maintenance of the looping ratio (Fig 2L). This supports the model that looping morphogenesis is completed by 2 days post fertilisation (dpf) [28] and the chambers subsequently undergo other morphological rearrangements to shape the tissue.

Ventral and lateral views of the heart suggest that chamber orientation changes during morphogenesis, in line with previously described chamber realignments [28,30]. To characterise how distinct morphogenetic processes in individual chambers contribute to heart development, *morphoHeart* was developed with the functionality to divide meshes into sections through a user-defined plane, in this instance cutting through the atrioventricular canal (Fig 2M). Division of the meshes into individual chambers, followed by the definition of each chamber's pole and apex, allows a more in-depth quantitative analysis of relative changes in chamber orientation during morphogenesis. The orientation of each chamber relative to a reference vector can be calculated from both a ventral view visualising how chambers align alongside each other (S2E and S2F Fig), and from a lateral view visualising how chambers rotate relative to each other around the AVC (S2J and S2K Fig). As the early heart tube undergoes looping, the ventricle pivots towards the right of the embryo or heart midline and the angle between both chambers reduces (S2H and S2I Fig). This rearrangement may facilitate concomitant growth of both chambers and looping of the heart. After looping has finished, the ventricle has pivoted back to towards the left side of the embryo causing the angle between chambers to increase again (S2I Fig). Analysis of chamber orientation from the lateral view

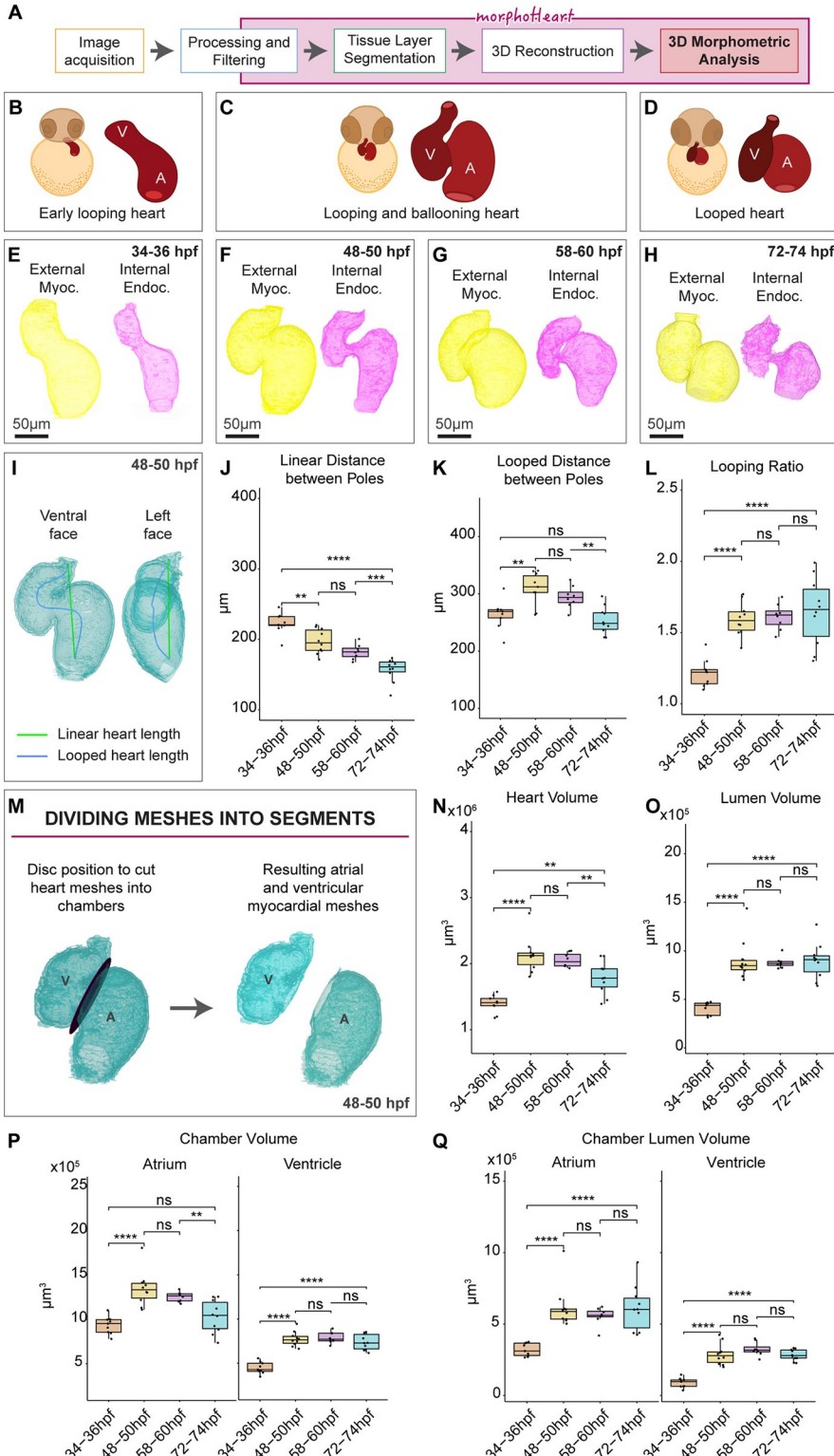

**Fig 2. The zebrafish heart grows and compacts during early cardiac morphogenesis.** (A) Flow diagram describing the phases involved in the process of acquiring comprehensive 3D morphometric data using *morphoHeart*. (B–D) Schematic depicting the early stages of heart development analysed, including looping of the early tube (B, 34–36 hpf), looping and ballooning (C, 48–60 hpf), and the looped heart (D, 72–74 hpf). (E–H) Reconstructions of myocardial and endocardial meshes during heart development. (I–L) Analysis of heart looping. Linear heart length (green line) and

heart centreline or looped heart length (blue line) are extracted and measured (I). As the heart develops, the poles move closer together (J). During looping morphogenesis, the centreline's looped distance elongates between 34 and 50 hpf, and subsequently shortens (K). Looping ratio also increases between 34 and 50 hpf, but then remains constant (L). (M) Cardiac chambers can be separated via placement of a user-defined disc. (N, O) Quantification of total heart volume reveals the heart increases in volume between 34 and 50 hpf and compacts again by 72 and 74 hpf (N). Lumen volume increases with heart volume and is then maintained (O). (P, Q) Analysis of chamber volume reveals while both chambers grow between 34 and 50 hpf, ventricle volume is maintained while the atrium shrinks (P). Lumen size in both chambers is maintained post-48 hpf (Q). One-way ANOVA with multiple comparisons.* $p < 0.05$, ** $p < 0.01$, *** $p < 0.001$, **** $p < 0.0001$, ns = not significant; 34–36 hpf: $n = 9$; 48–50 hpf: $n = 10$; 58–60 hpf: $n = 8$; 72–74 hpf: $n = 10$. Plots display median and quartiles. The numerical data underlying this figure can be found in S1 Data.

demonstrates that again the position of the atrium remains relatively static, while the ventricle undergoes a rotation to initially become aligned more parallel with the linear plane of the heart, which is subsequently reversed post-looping (S2L–S2N Fig).

Visual inspection of the heart meshes suggests that the heart grows and shrinks between 34 and 72 hpf (Fig 2E–2H). To confirm this, volumetric measurements of the external myocardium (as a proxy for whole heart size) and internal endocardium (as a proxy for lumen volume) were analysed. As the heart transitions from an early looping tube to a looped structure, the total volume of the whole organ, including the lumen, significantly increases (Fig 2N and 2O), expanding the blood-filling capacity of both chambers. Surprisingly, despite this growth, between 58 and 74 hpf heart volume significantly reduces (Fig 2N). However, regardless of this reduction in heart volume, lumen capacity is maintained from 48 hpf (Fig 2O), suggesting that remodelling of cardiac tissue occurs during early maturation, but this does not impact cardiac capacity. Individual analysis of chamber size reveals that both chambers grow between 34 and 50 hpf (Fig 2P), accompanied by a substantial increase in lumen volume (Fig 2Q). However, once the heart has undergone looping, the chambers display different dynamics, with the atrium reducing significantly in volume between 58 and 74 hpf, while ventricular volume is maintained (Fig 2P). Similar to analysis of the heart's size as a whole, lumen size of both chambers is maintained from 48 hpf (Fig 2Q), suggesting that the reduction in atrial volume is not due to a shrinkage of the whole chamber, but may instead represent a reduction in the amount of tissue comprising this chamber.

## The cardiac chambers undergo distinct geometric changes during heart looping and chamber expansion

The atrial and ventricular chambers eventually adopt very different morphologies in the mature heart [31,32]. To understand the temporal geometric changes of the developing chambers, atrial and ventricular shape were quantified by measuring the dimensions of the ellipsoid that best fits the chamber myocardial mesh (Figs 3A, S3A and S3B), including chamber width, length, depth, and asphericity (deviation of an ellipsoid from a sphere). During looping, the atrium maintains its length and width, while increasing in depth (S3C–S3E Fig), suggesting this elongation in this axis is responsible for the significant increase in atrial size, potentially representing a ballooning-type growth. Meanwhile, the ventricle lengthens (S3F–S3H Fig), suggesting that enlargement of the ventricle during looping is due to chamber elongation. Once the heart has looped at 48 to 50 hpf, the reduction in atrial volume we observed between 48 and 74 hpf (Fig 2P) is accompanied by a gradual increase in depth and significant shortening of the atrium (S3C–S3E Fig), resulting in rapid adoption of a more spherical morphology by 72 to 74 hpf (Fig 3B). In contrast to the atrium between 48 and 74 hpf, the ventricle only increases in depth (S3F–S3H Fig), maintaining its bean-shaped morphology (Fig 3C).

Cardiac ballooning is a conserved process by which the chambers emerge from the linear heart tube [33,34]. In zebrafish, expansion of the chamber regions of the heart tube, a process

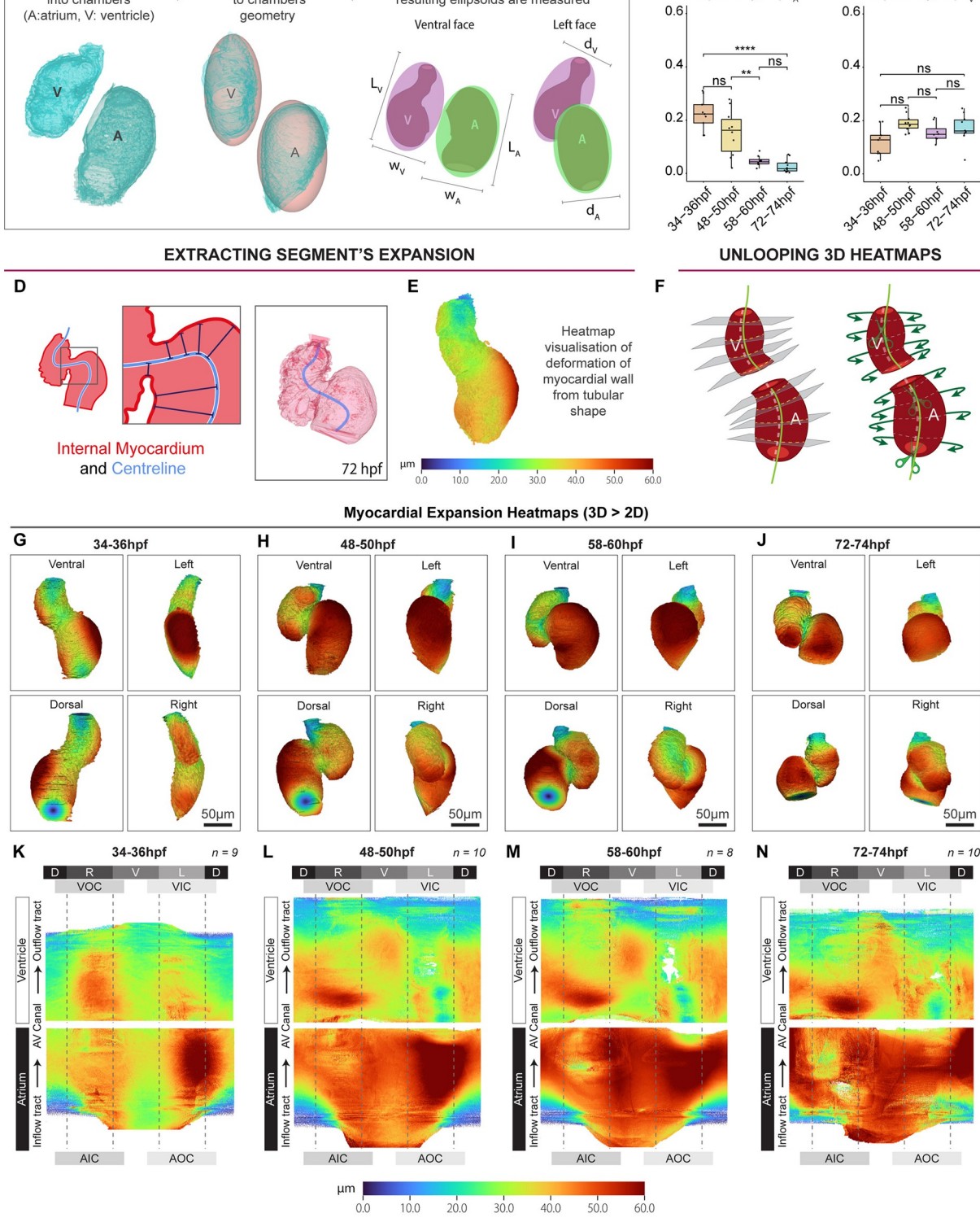

**Fig 3. Visualisation and quantification of chamber deformation reveals chamber-specific differences in growth.** (A–C) Ellipsoids are fitted to chambers to quantify chamber geometry (A). The atrium becomes more spherical (asphericity tends to 0) during development (B), while ventricle asphericity remains stable (C). One-way ANOVA with multiple comparisons. * $p < 0.05$, ** $p < 0.01$, *** $p < 0.001$, **** $P < 0.0001$. (D–F) Myocardial expansion/deformation can be quantified by measuring the distance between the myocardial centreline and the internal or external myocardial mesh (D). This value can then be mapped onto the internal or external myocardial mesh using a heatmap to

visualise 3D cardiac ballooning (E). D and E depict this process using the internal myocardial mesh. 3D heatmaps can be unrolled into a standard 2D geometry for aggregation and comparison (F). (G–J) Visualisation of 3D myocardial ballooning heatmaps mapped to the external myocardial mesh identifies substantial deformation of the atrial outer curvature at 34–36 hpf (G). By 48–50 hpf, this outer curvature deformation is enhanced, and the atrium is more ballooned than the ventricle (H). The ventricular apex can be seen emerging (H–J). (K–N) Unrolled 2D ballooning heatmaps allows averaging of multiple hearts to identify conserved regions of deformation. By 74 hpf deformation of the atrium has become more uniform (N). Labels around the 2D heatmaps indicate cardiac region: D—dorsal, V—ventral, L—left, R—right, AOC—atrial outer curvature, AIC—atrial inner curvature, VOC—ventricular outer curvature, VIC—ventricular inner curvature, AV Canal—atrioventricular canal; 34–36 hpf: *n* = 9; 48–50 hpf: *n* = 10; 58–60 hpf: *n* = 8; 72–74 hpf: *n* = 10. Plots display median and quartiles. The numerical data underlying this figure can be found in S1 Data.

akin to ballooning, has been described as occurring concomitant with cardiac looping from around 34 hpf [35,36], with the 2 processes together shaping the heart. To visualise and quantify regional expansion of the heart tube, the shorter distance between the heart's centreline and the internal or external myocardial mesh was calculated throughout the heart (Fig 3D), and mapped onto either myocardial mesh using a colour-coded representation of chamber expansion (Fig 3E and 3G–3J). This analysis reveals the emergence of the outer curvature of the atrium at 34 hpf (Fig 3G) which becomes more pronounced as looping progresses (Fig 3H), when expansion of the ventricular outer curvature is also initiated.

While analysis of individual 3D hearts is valuable for visualising localised regions of chamber expansion, it makes comparative analysis between biological replicates or stages relatively subjective. To address this, we incorporated a method into *morphoHeart* to convert 3D heatmaps into 2D matrices which can be averaged. This involves "cutting" each chamber's 3D heatmap along a dorsally projected centreline, then "unrolling" and "straightening" the cut surface around the extended centreline, resulting in a planar projection of the 3D heatmaps (Figs 3F, S4A and S4B). Multiple samples with the same 2D format can then be combined to generate an average heatmap (Figs 3K–3N, S4C and S4D), which represents an average metric (for example, chamber expansion), from multiple replicates, in which conserved features can be traced back to specific regions within the organ (S4E and S4F Fig). Analysis of 2D ballooning heatmaps confirms that by 74 hpf deformation of the atrium has become more uniform, representing the increase in sphericity we previously identified (Fig 3B), while the localised expansion of the ventricular apex becomes gradually more pronounced (Fig 3K–3N).

This analysis of heart development uses live *Tg(myl7:lifeActGFP);Tg(fli1a:AC-TagRFP)* embryos; however, we wanted to examine the range of applications of *morphoHeart* depending on the researchers' individual needs and transgenic lines available to them.

We first investigated whether choice of transgenic line impacted quantitative analysis of the heart and/or myocardium. We compared quantitative analysis of hearts in live *Tg(myl7:lifeActGFP)* transgenic embryos to *Tg(myl7:BFP-CAAX)* embryos at 48 to 50 hpf (in which myocardial membranes are labelled with BFP), as well as a new *Tg(myl7:Citrine)* transgenic line at 72 to 74 hpf (in which myocardial cells are labelled with a non-localised Citrine).

When comparing *Tg(myl7:BFP-CAAX)* hearts with *Tg(myl7:lifeActGFP)* hearts at 48 to 50 hpf, we observe subtle volumetric differences. The ventricles appear smaller (evident through both quantitative analysis, and ballooning heatmap analysis), which does not appear to be driven by a reduction in myocardial volume, but rather less expansion of the chamber (S5A, S5B, and S5E Fig). We also observe that while the atria are the same size in general between the 2 lines, there is less myocardial tissue in *Tg(myl7:BFP-CAAX)* atria (S5F Fig). Analysis of myocardial thickness 3D heatmaps demonstrates that this is due to poor signal in the dorsal (back) face of the atrium (S5D Fig, yellow arrow), and thus some "loss" of atrial tissue compared to the *Tg(myl7:lifeActGFP)* line. Looping ratio is unaffected, suggesting looping itself is independent of the transgene (S5G Fig).

When comparing *Tg(myl7:Citrine)* hearts with *Tg(myl7:lifeActGFP)* hearts at 72 to 74 hpf, we again observe some differences in volumes, although not the same as those observed in the *Tg(myl7:BFP-CAAX)* line. We find that overall atrial size is larger in *Tg(myl7:Citrine)* transgenics compared to the *Tg(myl7:lifeActGFP)* line, while ventricular size is smaller (S5H–S5J Fig). This larger atrial size is due to a more advanced growth of the chamber (as evidenced by greater chamber ballooning and coincident greater atrial myocardial volume), whereas the smaller ventricular size appears to be the result of less ballooning of the ventricle (S5I and S5K Fig). Again, looping ratio is not affected (S5L Fig).

Next, we compared *morphoHeart* analysis of the heart and myocardium in fixed *Tg(myl7: lifeActGFP)* to the data we acquired from live images to confirm the limitation that tissue fixation affects morphometric analyses. We found that fixation leads to a reduction in both whole heart and chamber-specific volumetric measurements. This included a reduction in both the size of the chambers and profound shrinking of the myocardial tissue (S6A–S6F Fig). Additionally, ballooning heatmap analysis also demonstrates that fixation causes folding in the thin atrial myocardium (S6B Fig, arrow). However, it is worth noting that some morphological measurements such as looping ratio do not appear to be affected by fixation (S6G Fig); therefore, fixed tissue imaging may still allow some accurate morphological analysis. Together, this highlights the power of *morphoHeart* to identify small conserved differences in heart morphology or growth during development and between different populations.

## Individual cardiac chambers undergo separate processes of tissue growth and regional shrinkage

*morphoHeart* analyses of heart development highlight that cardiac chambers undergo geometric changes commensurate with ballooning; however, chamber size dynamics after initial looping indicate chamber expansion may not be a process only of tissue growth. We therefore investigated cardiac tissue volume in more detail (Fig 4A–4E). Total myocardial volume increases as the heart undergoes looping morphogenesis, but surprisingly reduces again between 48 and 60 hpf (Fig 4B). Analysis of the tissue volume of individual chambers reveals distinct chamber-specific myocardial tissue dynamics that account for this observation. Atrial myocardial tissue volume remains relatively consistent as the heart undergoes initial looping, but significantly reduces as the heart matures (Fig 4C). Conversely, ventricular myocardial volume increases significantly as the heart undergoes looping morphogenesis (Fig 4C), in line with the addition of second heart field cells to the arterial pole between 24 hpf and 48 hpf [37,38], and subsequently remains constant. Analysis of endocardial tissue dynamics reveals a decrease in endocardial volume in the whole heart between 58 and 74 hpf (Fig 4D) in both chambers (Fig 4E), which may reflect the general reduction in cardiac size at the later stage.

This reduction in atrial tissue volume after looping morphogenesis is complete (at 48 to 50 hpf) is in line with our observation that total heart volume decreases over the same time frame driven primarily by a reduction in atrial size while lumen size is maintained (Fig 2). These changes in tissue volume could result from a reduction in number of cells or a reduction in cell size. To address both these questions, we developed *morphoCell*, an integrated module in *morphoHeart* to perform cell analysis (S7 Fig). We imaged *Tg(myl7:BFP-CAAX);Tg(myl7:H2B-mScarlet)* double transgenic embryos in which cardiomyocyte nuclei are labelled (Fig 4F), and used Imaris software to extract nuclei coordinates. These, together with the myocardial *z*-stack were given as input into *morphoCell* where a plane was defined to separate chambers and allocate nuclei as atrial or ventricular (S7C Fig). This demonstrated that total cardiomyocyte number in the heart increases between 48 and 60 hpf (Fig 4G), driven mainly by the ventricle (Fig 4H), likely through continued addition and differentiation of second heart field (SHF) cells

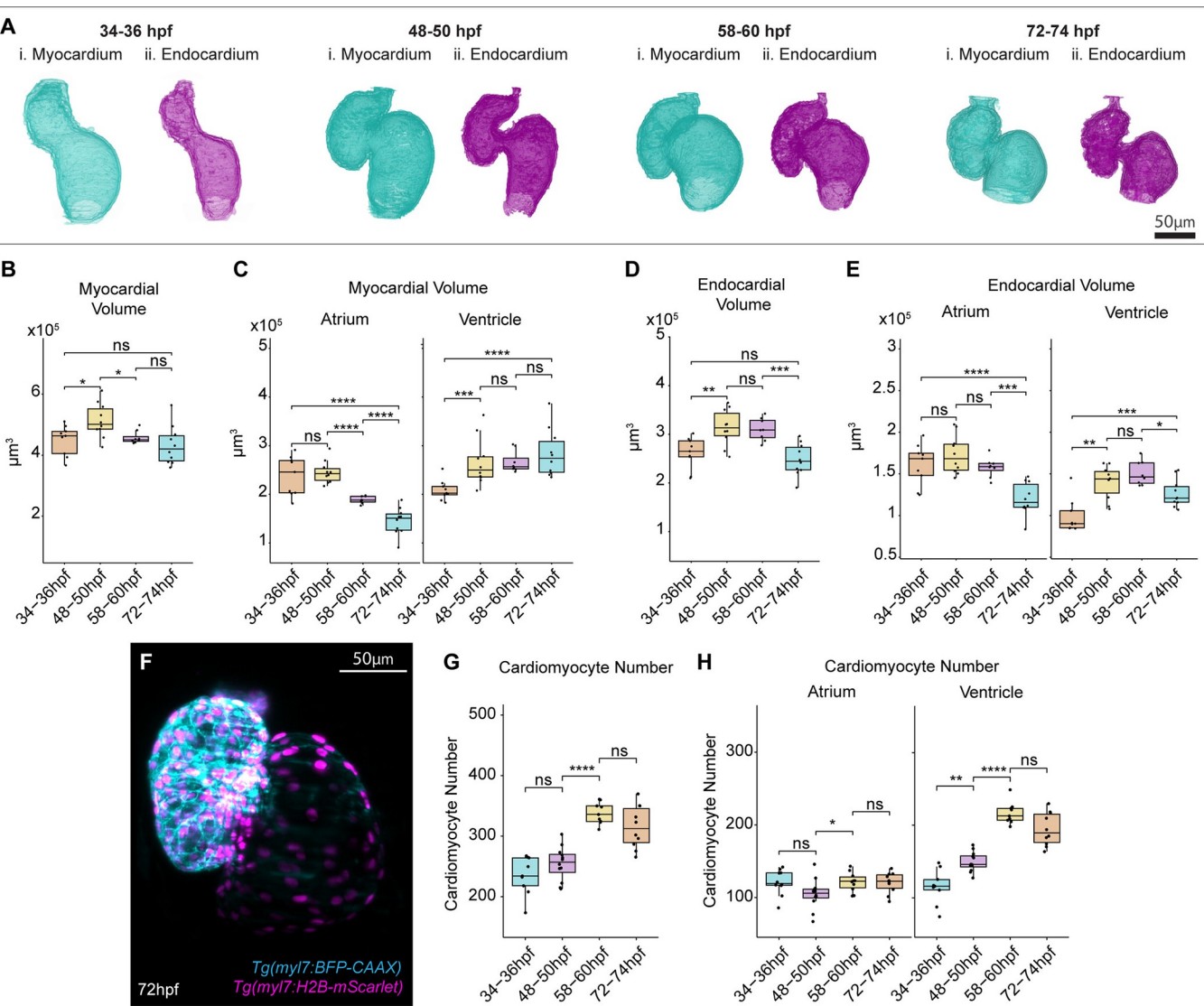

**Fig 4. The atrium and ventricle exhibit different tissue dynamics during morphogenesis.** (A–C) Quantification of myocardial tissue volume from myocardial meshes (A). Total myocardial volume increases during looping and reduces at early stages of maturation (B). Chamber-specific analysis reveals a later reduction in atrial myocardium volume compared with an earlier increase and maintenance in ventricular myocardium volume (C). (D, E) Quantification of endocardial tissue volume from endocardial meshes. Total endocardial volume decreases after 58 hpf (D), driven by a reduction in endocardial tissue in both the atrium and ventricle (E); 34–36 hpf: $n = 9$; 48–50 hpf: $n = 10$; 58–60 hpf: $n = 8$; 72–74 hpf: $n = 10$. (F–H) Quantification of cardiomyocyte number, from live lightsheet $z$-stack images of *Tg(myl7:BFP-CAAX);Tg(myl7:H2B-mScarlet)* (F). The total number of cardiomyocytes increases between 48 and 60 hpf (G). Atrial cardiomyocyte number remains mostly constant, while ventricular cardiomyocyte number increases (H). One-way ANOVA with multiple comparisons. * $p < 0.05$, ** $p < 0.01$, *** $p < 0.001$, **** $p < 0.0001$, ns = not significant; 34–36 hpf: $n = 8$; 48–50 hpf: $n = 10$; 58–60 hpf: $n = 7$; 72–74 hpf: $n = 10$. Plots display median and quartiles. The numerical data underlying this figure can be found in S1 Data.

[38]. However, cell number in the atrium remained mostly constant, suggesting the reduction in atrial myocardial volume is not driven by cell loss or cell death, and together with previous studies [37,39,40] suggests that SHF addition to the venous pole predominantly occurs prior to 34 hpf.

We therefore investigated whether atrial cardiomyocyte size reduces after initial heart looping morphogenesis. *morphoCell* can assign cardiomyocyte nuclei into clusters and measure 3D cardiomyocyte internuclear distance (IND) as a proxy for cell size (S7D–S7F Fig). IND analysis of total chamber cardiomyocytes revealed that atrial cardiomyocytes slightly increase in size

during looping, but once looping has occurred only ventricular cardiomyocytes reduce in size (Fig 5A). Early chamber morphogenesis involves regionalised and chamber-specific changes in tissue morphology [36], and therefore we wished to assess cardiomyocyte size in more detail. Chamber-specific nuclei can be assigned to discrete regions of the chamber, such as the inner or outer curvatures, or dorsal or ventral face (Fig 5B). This revealed regional differences in atrial cardiomyocyte size dynamics, with ventral and outer curvature cardiomyocytes associated with atrial expansion, and ventral atrial cardiomyocytes specifically reducing in size after looping (Fig 5C). Similarly, ventricular cardiomyocytes exhibit regional differences in expansion and reduction, with inner curvature cardiomyocyte size remaining relatively stable, while cardiomyocytes on the ventral, outer, and dorsal faces of the ventricle undergo more dynamic changes in size throughout development (Fig 5D).

Reduction in internuclear distance may reflect a change in cell geometry rather than a reduction in cell volume (i.e., cells get taller and narrower). Similarly, the decrease in atrial myocardial volume observed (Fig 4) is unlikely to be only attributed to a relatively modest and regional reduction in atrial cardiomyocyte size. We therefore sought to visualise myocardial wall thickness during development. As each tissue mesh comprises an outer and inner mesh, the shorter distance between these 2 meshes can be measured across the myocardial tissue (Fig 5E) and mapped onto the myocardial mesh as a heatmap (Fig 5F), providing a visual readout of myocardial thickness across development (Fig 5G–5J). Analysis of 2D unrolled and averaged heatmaps demonstrates that the ventricular myocardial wall is consistently thicker than the atrial wall (Fig 5K–5N). Importantly, the atrial wall thins over development, which together with the regional reduction in cardiomyocyte IND supports the hypothesis that cardiomyocytes shrink after initial looping morphogenesis. Together this suggests that the chamber-specific changes in geometry and size that occur post-looping may be driven by regionalised changes in cell geometry and volume.

Our *morphoHeart* volumetric analysis thus suggests that concomitant with heart looping, the heart grows significantly through increase in cardiomyocyte size, accrual of cardiomyocytes and expansion of both chamber lumens until around 50 hpf. Subsequently, cardiomyocytes undergo chamber-specific regional shrinkage while the lumen of the tissue is maintained, facilitating geometric changes that result in the adoption of specific ballooned morphologies in each chamber while maintaining cardiac capacity.

## The cardiac ECM undergoes dynamic regionalised and chamber-specific volumetric remodelling

We have previously shown that the cardiac ECM is regionalised prior to the onset of heart looping, where the atrial ECM is thicker than the ventricular ECM, and the left atrial ECM is expanded compared with the right [7]. We wished to investigate whether this regionalisation of ECM is maintained throughout early heart development and how it relates to cardiac morphogenesis. We aimed to perform this analysis in live embryos (avoiding alteration of tissue morphology or matrix composition that may be introduced through fixation, dehydration, or processing), and without the use of ECM sensors (such as the previously published HA sensor [6,41]), to avoid assumptions of ECM content. We took advantage of the contour libraries generated by *morphoHeart* to segment the negative space between the internal myocardial contour and external endocardial contour (Fig 6A), generating a mesh representing the cardiac ECM. Visualisation of cardiac ECM meshes across cardiac development (Fig 6B) revealed some expected features, such as a patchy reduction of the ECM in the ventricle between 48 and 74 hpf, in line with previous reports in both mouse and zebrafish that ECM degradation occurs at the onset of ventricular trabeculation [42,43]. Quantitative analysis revealed that the ECM is

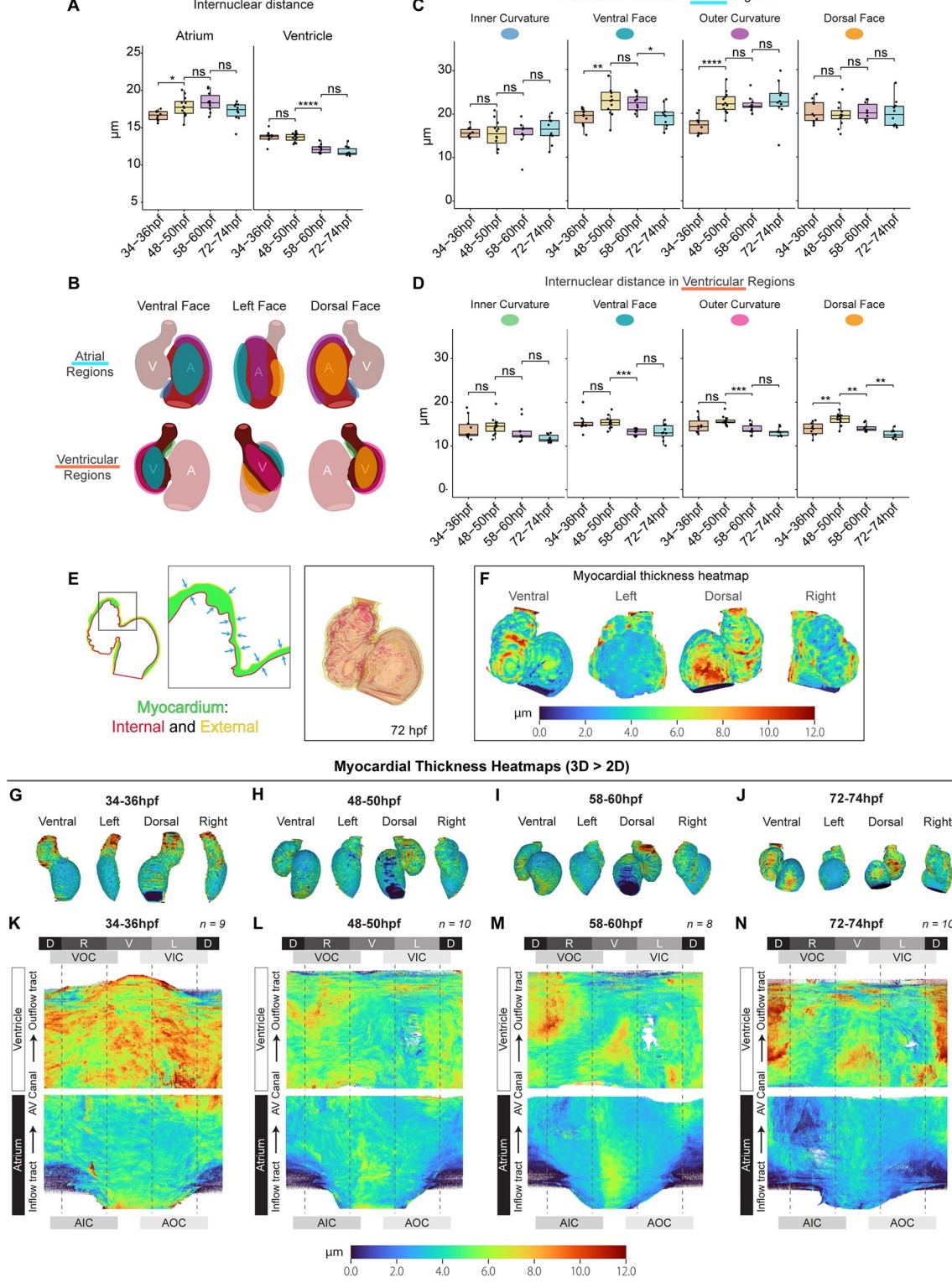

**Fig 5. Cardiac chambers undergo regionalised reduction in cell size.** (A–D) Quantification of internuclear cardiomyocyte distance as a proxy for cell size reveals an early increase in atrial cardiomyocyte size and a later reduction in ventricular cardiomyocyte size. Each chamber is subdivided into regions (distinguished by different colours) for more granular analysis (B–D). Growth and decrease in atrial cardiomyocyte size occurs predominantly in ventral and outer curvatures (C). Ventricular dorsal cardiomyocytes expand early, and all ventricular cardiomyocytes apart from those on the inner curvature subsequently decrease in size (D). Each dot

represents the average internuclear distance, per region, in one heart. One-way ANOVA with multiple comparisons. * $p < 0.05$, ** $p < 0.01$, *** $p < 0.001$, **** $p < 0.0001$; 34–36 hpf: $n = 10$; 48–50 hpf: $n = 12$; 58–60 hpf: $n = 10$; 72–74 hpf: $n = 10$. (E–N) Myocardial wall thickness is quantified by measuring the distance between the inner and outer myocardial meshes (E) and mapped onto the outer myocardial mesh using a heatmap to visualise myocardial thickness in 3D (F). 3D myocardial thickness heatmaps (G–J) are unrolled to 2D, and average 2D heatmaps generated for each time point (K–N), illustrating that the atrial wall is consistently thinner than the ventricular, and that both chamber walls thin as development progresses. Labels around the outside indicate cardiac region: D— dorsal, V—ventral, L—left, R—right, AOC—atrial outer curvature, AIC—atrial inner curvature, VOC—ventricular outer curvature, VIC—ventricular inner curvature, AV Canal—atrioventricular canal. Plots display median and quartiles. The numerical data underlying this figure can be found in S1 Data.

highly dynamic, first significantly expanding between 34 and 50 hpf as the heart loops, before reducing between 58 and 74 hpf as the heart matures (Fig 6C). Chamber-specific analysis revealed that atrial ECM volume is consistently higher than ventricular ECM volume, and while both chambers exhibit similar dynamics, the timing is different, with the ventricular ECM volume reducing between 48 and 60 hpf, earlier than the atrial ECM volume which reduces only between 58 and 74 hpf (Fig 6D). This suggests that the chambers have distinct mechanisms for managing ECM degradation or reduction.

To visualise ECM thickness, we used *morphoHeart* to measure the distance between the ECM mesh contours (Fig 6E and 6F) and mapped the thickness values onto the external ECM tissue contour, using a heatmap scale to visualise ECM thickness (Fig 6G). Inspection of 3D ECM heatmaps throughout development reveals that the cardiac ECM is highly regionalised, with an expansion of the ECM on the outer curvature of the atrium at 34 to 60 hpf (Fig 6H– 6J), which is repositioned to the dorsal atrial face by 74 hpf (Fig 6K). 2D unrolled and averaged heatmaps facilitated a granular analysis of ECM regionalisation. At 34 hpf, the ECM is thickest on the left (outer) curvature of the atrium, in line with our previous findings at 26 hpf [7]. We also observed a mild localised ECM thickening on the right (inner) curvature of the atrium and that the left-sided ECM thickening expands into the proximal ventricle (Fig 6L). At 48 to 60 hpf, this regionalised thickening of the ECM in the atrium is maintained, and the magnitude increased, on both outer and inner curvatures (Fig 6M and 6N), while the ECM thickening in the inner curvature of the ventricle is slightly reduced at 60 hpf (Fig 6N). By 72 hpf, the atrial ECM is still regionally expanded, but the left-sided expansion has reduced in magnitude and is now positioned to the dorsal face (Fig 6O). The left-sided inner ventricular ECM expansion has reduced, although the ECM in that region still appears slightly thicker than the right side (outer curvature), which may be in line with regionalisation of trabeculation onset [43]. To quantify ECM volume specifically in these chamber regions, *morphoHeart* used the centreline to divide each chamber into left and right sides (Fig 6P). This confirmed that the atrial ECM is greater on the left side and expands more significantly to amplify the magnitude of the asymmetry as the heart undergoes morphogenesis (Fig 6Q). While left and right ventricular ECMs are more similar in volume, the left side undergoes a more dynamic expansion and reduction (Fig 6R), in line with the changes in thickness depicted in the heatmaps.

To validate our negative-space ECM segmentation findings, we conducted additional analysis using *Tg(myl7:BFP-CAAX);Tg(ubi:ssNcan-EGFP)* embryos, in which HA throughout the embryo is labelled with a GFP biosensor. This allowed us to specifically visualise HA distribution and confirm the ECM distribution patterns observed with *morphoHeart* (S8 Fig). Segmentation of the GFP channel was particularly challenging (S8A–S8B' Fig, see Discussion), but the HA sensor experiments revealed similar regionalisation patterns of the ECM, particularly highlighting the expanded ECM on the outer curvature of the atrium during early heart development (S8D–S8J Fig) corroborating our previous findings. Together, *morphoHeart* reveals novel chamber-specific dynamics in ECM expansion and reduction during cardiac morphogenesis.

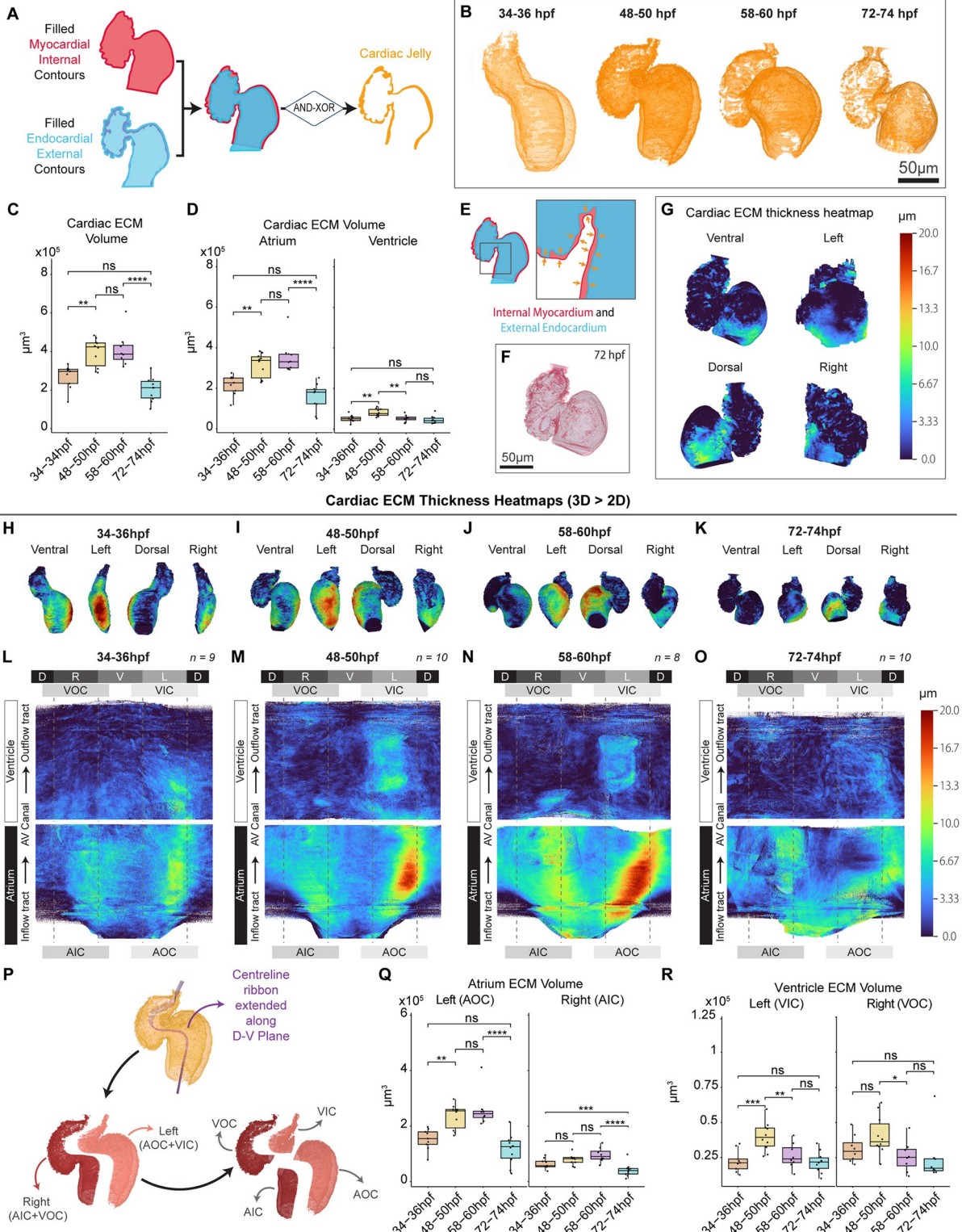

**Fig 6. The ECM undergoes chamber-specific regionalised expansion and reduction during heart morphogenesis.** (A) Schematic depicting the approach used to generate cardiac ECM meshes, by combining the filled external endocardial contour and the filled internal myocardial contour using AND and XOR logical operations. (B–D) Volumetric 3D reconstructions of the cardiac ECM during heart development (B), showing an apparent reduction in the ventricular ECM at 72–74 hpf. Quantification of total cardiac ECM volume reveals a significant increase in ECM volume between 34 hpf and 50 hpf, followed by a reduction between 58 hpf and 74 hpf (C). The majority of

cardiac ECM is found in the atrium, and while both chambers expand their ECM during looping, ventricular ECM reduces first between 48 hpf and 60 hpf, while atrial ECM is reduced only after 58 hpf (D). (E–G) ECM thickness is quantified by measuring the distance between the outer endocardial mesh and inner myocardial mesh (E) and mapped onto the inner myocardial mesh (F) using a heatmap to visualise ECM thickness in 3D (G). (H–O) 3D heatmaps reveals the cardiac ECM is thicker in specific regions of the heart (H–K). Unrolled and averaged 2D ECM thickness heatmaps reveals the ECM is thicker in the atrium than the ventricle and in particular in the outer curvature of the atrium at 34–60 hpf (L–N). The atrial ECM is still regionalised at 72–74 hpf but the thickening is repositioned to the dorsal face of the atrium (O). Labels around the outside indicate cardiac region: D—dorsal, V—ventral, L—left, R—right, AOC—atrial outer curvature, AIC—atrial inner curvature, VOC—ventricular outer curvature, VIC—ventricular inner curvature, AV Canal—atrioventricular canal. (P) Schematic illustrating the cutting of the ECM mesh into left and right regions for both the atrium and ventricle. (Q, R) Quantification of ECM volume in outer and inner curvatures of the atrium (Q) and ventricle (R) reveal the regionalised dynamics that drive cardiac ECM expansion and reduction. One-way ANOVA with multiple comparisons. * $p < 0.05$, ** $p < 0.01$, *** $p < 0.001$, **** $p < 0.0001$, ns = not significant; 34–36 hpf: $n = 9$; 48–50 hpf: $n = 10$; 58–60 hpf: $n = 8$; 72–74 hpf: $n = 10$. Plots display median and quartiles. The numerical data underlying this figure can be found in S1 Data.

### ECM crosslinker Hapln1a promotes cardiac growth dynamics

We previously demonstrated that the ECM crosslinker Hapln1a is required for regulating early ECM volume asymmetries and heart growth [7]. To validate that *morphoHeart* can deliver more detailed analyses of mutant phenotypes, we performed *morphoHeart* analysis of *hapln1a* mutants at 34 to 36 hpf, 48 to 50 hpf, and 72 to 74 hpf (Fig 7A and 7B). Analysis of heart size revealed that *hapln1a* mutant hearts are only significantly smaller than siblings at 48 hpf (Fig 7C), once the heart has undergone morphogenesis. This reduction in heart size is driven by a failure of the *hapln1a* mutant atrium to expand by 48 hpf (Fig 7D–7G). Analysis of both lumen and myocardial volume reveals that defective atrial growth is driven by limited expansion of the *hapln1a* mutant lumen (Fig 7H–7L), demonstrating that Hapln1a links atrial ballooning to lumen expansion.

Finally, we examined ECM volume and distribution in *hapln1a* mutants. The volume of cardiac ECM per chamber region is reduced (Fig 7M and 7N), resulting in a diminished contribution of the cardiac ECM to the total heart volume (S9A–S9C Fig). We observed that *hapln1a* promotes the regionalised expansion of ECM between 34 and 50 hpf in both the atrium and ventricle (Fig 7M and 7N), as well as apparently protecting the ECM at both the inner curvature and outer of the ventricle from premature reduction, degradation, or compaction by 74 hpf (Fig 7N). Surprisingly, despite this significant reduction in regional ECM volume in the *hapln1a* mutant, averaged 2D ECM thickness heatmap analysis of *hapln1a* mutants at between 34 hpf and 50 hpf reveals some residual regionalisation and asymmetric expansion (S9F and S9G Fig), although the magnitude and expanse of this left thickening appears reduced compared to wild type (S9D and S9E Fig), in line with quantitative analysis (Fig 7M).

Together, these findings suggest that *hapln1a* plays a critical role in amplifying the magnitude of ECM asymmetries during cardiac development. The observed retention of ECM expansion in the atrial outer curvature of *hapln1a* mutants, despite the overall reduction, indicates that while *hapln1a* significantly contributes to ECM regionalisation, it is not the sole determinant. Other genes and molecular pathways are likely acting in concert with *hapln1a* to establish and maintain the complex regionalisation patterns within the cardiac ECM. This cooperative interaction among multiple factors is essential for promoting proper atrial expansion and overall cardiac morphogenesis. Understanding these interactions will be crucial for elucidating the full regulatory network governing ECM dynamics during heart development.

## Discussion

### *morphoHeart* reveals new insights into cardiac morphogenesis

Early heart morphogenesis is a complex asymmetric process that requires the timely coordination of distinct events, including looping and regional ballooning of the linear heart tube.

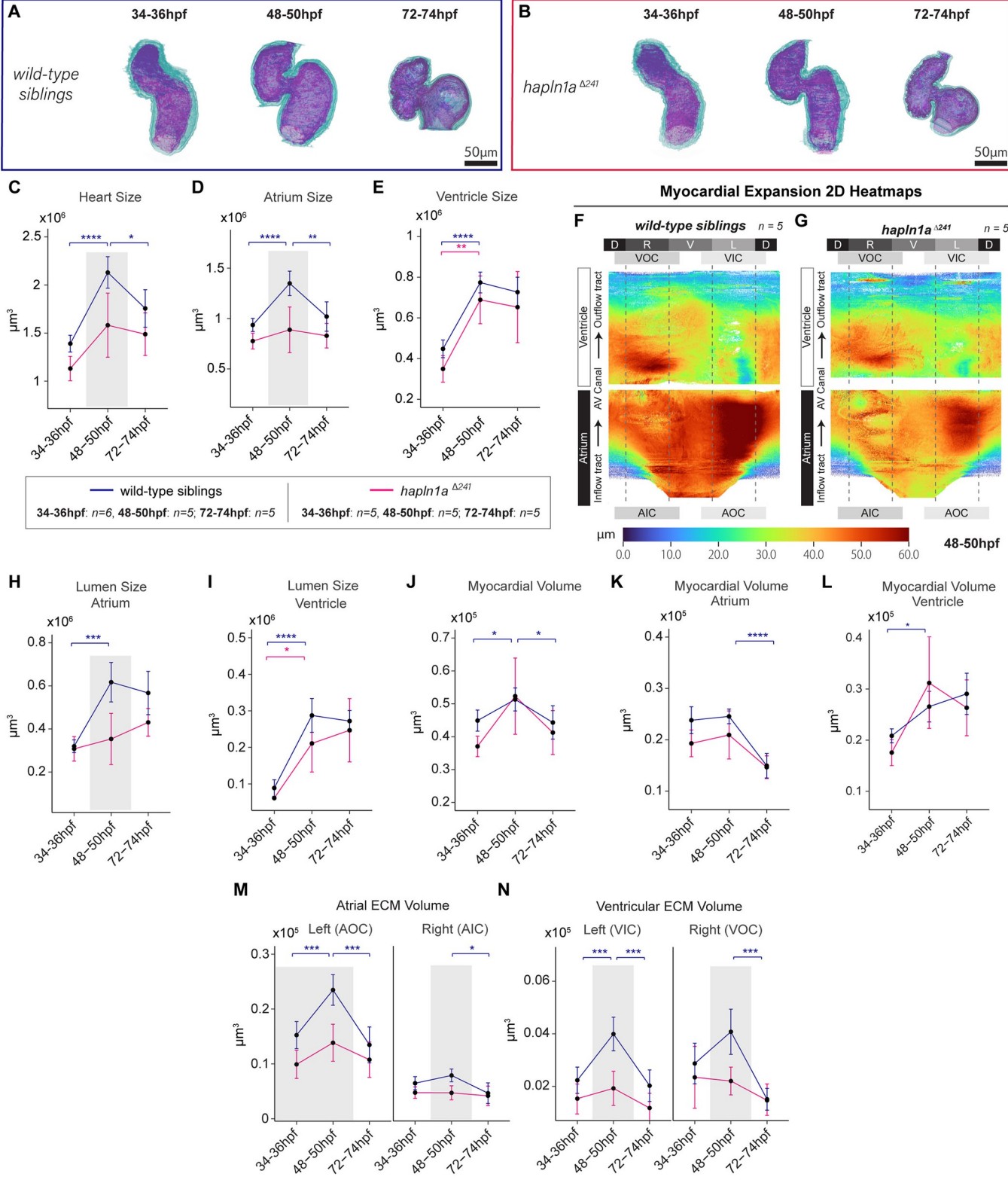

**Fig 7. *hapln1a* mutants exhibit defects in atrial growth and ECM expansion.** (A, B) Myocardial (green) and endocardial (magenta) 3D reconstructions of wild-type sibling (A) and *hapln1a* mutant hearts (B) at 34–36 hpf, 48–50 hpf, and 72–74 hpf. (C–E) Quantification of heart size reveals that *hapln1a* mutant hearts (pink) are smaller at 48–50 hpf than wild-type siblings (blue, C), largely due to a failure of the atrium to balloon by 48–50 hpf (D). (F, G) Averaged 2D chamber ballooning heatmaps of wild type (F, *n* = 5) and *hapln1a* mutant embryos (G, *n* = 5) at 48–50 hpf demonstrates that *hapln1a* mutant atria fail to

expand. (H–L) Quantification of lumen size and myocardial tissue volume shows that *hapln1a* mutants fail to expand the atrial lumen at 48–50 hpf (H), whereas the expansion dynamics of the ventricular lumen is unaffected (I). (M, N) Analysis of regional ECM volume in wild-type siblings and *hapln1a* mutants. ECM volume does not expand in either the atrium or ventricle of *hapn1a* mutants at 48–50 hpf compared to wild-type siblings (M, N). Asterisks indicate significant difference between time points for each genotype (blue indicates significance in siblings, pink indicates significance in *hapln1a* mutants). Grey boxes indicate significant difference between wild-type siblings and mutants at the indicated time point. Two-way ANOVA with multiple comparisons. * $p < 0.05$, ** $p$ 0.01, *** $p < 0.001$, **** $p < 0.0001$, ns = not significant. Wild type 34–36 hpf: $n = 6$; all other genotypes and stages: $n = 5$. Labels around the heatmap indicate cardiac region: D—dorsal, V—ventral, L—left, R—right, AOC—atrial outer curvature, AIC—atrial inner curvature, VOC—ventricular outer curvature, VIC—ventricular inner curvature, AV Canal—atrioventricular canal. Plots display median and quartiles. The numerical data underlying this figure can be found in S1 Data.

Using *morphoHeart*, we have demonstrated that the complex 3D morphological transformations of the zebrafish heart tube during cardiac development can be characterised through comparative integrated analysis of 3D morphometric parameters in wild-type hearts at key developmental stages.

*morphoHeart's* quantitative results of myocardial growth during early looping shows that the increase in myocardial mass is driven by growth of the ventricle, while the atrial myocardium remains constant, corroborating previous studies showing that second heart field addition occurs at the venous pole earlier than the arterial pole [37], likely prior to the stages we capture here. We further show for the first time an atrial-specific reduction in total myocardial volume after initial looping morphogenesis, while ventricular myocardial mass is maintained. *morphoHeart's* capability to perform integrated analyses demonstrate that this reduction in atrial myocardium, and maintenance in ventricular myocardium are both associated with regionalised reduction in cardiomyocyte size and changes in cell geometry, but in the ventricle myocardial volume is maintained through increased cell numbers. Chamber-specific analysis highlighted that these differences may be the result of ongoing chamber-specific refinement mechanisms; for example, the increase in ventricular cardiomyocytes could be due to ongoing addition of cells to the arterial pole from the SHF [37,38,40], and/or through the proliferation of cardiomyocytes during trabecular seeding [44]. Furthermore, the chamber-specific regional reductions in cell size we observe are in line with other studies that suggest that anisotropic cell shape changes drive tissue remodelling [28,36,45–48].

Our data suggests that movement of the ventricle primarily drives heart looping. We observe a combination of frontal and sagittal rotations in the ventricle suggesting that, contrary to the findings of previous studies [28], the deformation of the linear heart tube into an S-shaped loop does not solely take place in the frontal plane. Studies in other models have not only corroborated the three-dimensionality of looping morphogenesis process by describing the sequential frontal (left/right) and transverse (cranial/caudal) rotations of the chambers and OFT involved in looping morphogenesis [49–51], but also described the principal role played by the ventricle during this asymmetric process of looping [52,53]. This suggests that the ventricular rotations underpinning chamber rearrangements during cardiac looping morphogenesis are conserved across species.

## The cardiac ECM undergoes regionalised dynamic changes in volume

*morphoHeart* allows the first semi-automated 3D volumetric visualisation and analysis of the cardiac ECM in live embryonic hearts. We have shown that ECM expansion in both chambers is associated with the initial growth of the heart during looping and ballooning morphogenesis while ECM reduction, possibly driven by degradation, dehydration, or compaction, is subsequently linked to chamber-specific remodelling and maturation. Reduction in ECM volume occurs earlier in the ventricle than the atrium (between 48 and 50 hpf compared to 58 to 74 hpf), corresponding with the onset of ventricular trabeculation, which has been linked to

specific dynamics of ECM remodelling [42,43,54]. A recent study used manual segmentation of the *Tg(ubi:ssNcan-EGFP)* transgenic line at post-looping stages of development (50 to 96 hpf) to analyse chamber-specific ECM composition, however, did not identify the same reduction in ECM volume we observe in either the ventricle or atrium between 50 hpf and 72 hpf [55]. We speculate that these discrepancies in ECM dynamics may relate to differences in segmentation approaches, specifically the use of the HA-biosensor line, which we discuss further later.

ECM thickness analysis provides a more granular understanding of cardiac ECM distribution, including regional expansion of the ECM on the left side of the heart tube, in line with our previous observations [7] and previous studies describing the cardiac ECM of embryonic chick hearts as thicker on the left-right regions of the heart tube compared to the antero-posterior [56–58]. We also demonstrate for the first time that this regional expansion is maintained as the heart undergoes looping, chamber expansion, and early maturation, although in the atrium, this asymmetric expansion is relocated from the left face to the dorsal face as the heart matures.

Our analysis further revealed a distinct region of thickened ECM in the inner curvature of the atrium close to the venous pole, in the same area where previous studies have located the zebrafish pacemaker/sinoatrial node cells [37,59]. Studies in animal models have identified that the cells comprising the sinoatrial node are embedded within a biochemically and biomechanically distinct ECM that serves as a protective scaffold to pacemaker cardiomyocytes, reducing the mechanical strain and mechanotransduction they would experience from cardiac contractility [60], raising the possibility that these cells are similarly isolated in the zebrafish heart during development.

We reveal a novel requirement for Hapln1a in driving expansion of the atrial lumen, drawing parallels with studies in *Drosophila* demonstrating that proteoglycans and glycoproteins regulate expansion of the intestinal lumen [61], heart lumen [62], and interrhabdomeral space in the eye [63]. Importantly, our detailed 3D analysis of ECM thickness reveals new insights that were not possible with our previous cross-sectional analysis of the cardiac ECM [7]. First, we clearly show using statistical testing that Hapln1a drives the atrial ECM expansion that occurs between 34 hpf and 48 hpf, something we could not previously quantify. Second, contrary to our previous hypothesis, Hapln1a appears to be not the only factor driving regionalised expansion of the ECM since we can now appreciate that a small region of thicker ECM remains in the atrium of *hapln1a* mutants at early stages of development. Whether this is due to regionalised production of HA, proteoglycans, or other ECM crosslinking proteins, or rather is due to localised expression of ECM-degrading enzymes, is unclear. Third, we show that Hapln1a is required, directly or indirectly, for the more modest ECM regionalisation in the ventricle, and finally, we find that Hapln1a-driven ECM expansion at 48 to 50 hpf promotes expansion of the atrial lumen, which likely underlies the defective atrial ballooning in *hapln1a* mutants.

The role of Hapln1a in regulating ECM and atrial expansion likely stems from its function in the ECM. Hapln1a is a cross-linking protein that mediates the interaction between HA and proteoglycans [64–66], which in turn provides structure and biomechanical cues to tissues. Although proteoglycans and HA can interact and form complexes in the absence of link proteins, *in vitro* studies have shown that cross-linking allows the ECM to sustain higher loads (i.e., increased compressible resistance) while maintaining an elastic structure [66,67]. In addition, a stable and elaborate network of HA-PG complexes into which GAG chains can sequester water could result in formation of a hydrated and regionally expanded matrix that promotes atrial wall deformation, as well as signal transduction and proliferation through

mechanical tension which has been proposed to correlate endocardial growth with myocardial ballooning [68].

Together, our new findings showcase the superior power of *morphoHeart* for analysing ECM regionalisation and expansion and highlight the complexity of dynamic ECM composition in shaping the developing heart.

### *morphoHeart*, a new tool for morphometric analysis

To date, a single tool or pipeline cannot address all the processing and analytic requirements for analysis of 3D data sets. Fully automatic 3D segmentation of biological images can be computationally demanding and can perform poorly due to low local contrasts, high noise levels and signal from structures or artefacts surrounding the objects of interest [69,70]. Once the structures of interest have been segmented, either by fully automatic or manual segmentation, limited 3D object quantification in open-source software (e.g., 3D Viewer, MorphoLibJ, 3D Slicer) extracts few quantitative readouts, restricting the depth of the analysis. Some commercially available software specifically designed to analyse medical (e.g., Mimics Materialise, Belgium) or biological (e.g., arivis Vision 4D, Germany; Imaris, Oxford Instruments; Volocity, PerkinElmer) images provide a more extensive toolset for the morphological analysis of 3D objects. Nevertheless, the available metrics might not meet all the user-specific needs. *morphoHeart* is an exciting alternative offering a comprehensive pipeline that provides semi-automated segmentation capabilities and improves the suite of tools and quantifications available in the biological field to characterise organ morphology. While *morphoHeart* was initially developed for the study of zebrafish heart development, it can be applied to studying a wide range of morphogenetic processes in other organs and organisms, contributing to our understanding of the tissue organisation mechanisms and morphogenesis processes underpinning them.

In addition to the comprehensive morphometric quantifications performed by *morphoHeart*, it also provides the first negative-space segmentation of the cardiac ECM. A previous study used automated constrained mesh inflation and subtraction to define the negative space surrounding joints [71]; however, our contour-directed negative-space segmentation allows a more detailed analysis of regional expansions of the segmented matrix. While transgenic lines providing fluorescent readouts of HA have been described [6,41] which could be used for analysis of the cardiac ECM in *morphoHeart*, these lines are not specific to the heart, making clean segmentation challenging. Additionally, if ECM composition changes over time the use of ECM-specific sensor lines (which assume ECM content) could render inaccurate segmentations at stages where the chosen sensor is absent or minimally expressed, resulting in incomplete ECM volumes and leading to erroneous interpretations of ECM distribution. In contrast, *morphoHeart's* segmentation approach, which derives ECM volume from the negative space, is independent of ECM composition, allowing for a more consistent and unbiased analysis across developmental stages. To determine whether these different approaches for analysing ECM yield differing results, we compared HA-sensor segmentation to negative-space segmentation at 48 to 50 hpf and found that regionalisation of the ECM in the atrium and ventricle is observed regardless of the analysis method (S8 Fig). However, there are some differences between analysis methods, which we discuss later.

*morphoHeart's* GUI allows the morphometric analysis tools of *morphoHeart* to be easily applied to z-stack images of any fluorescently labelled tissue acquired by any imaging modality, enhancing user experience and broadening its applicability (Fig 8). As long as images are saved in TIF format and have good signal-to-noise ratio to facilitate automatic contour segmentation, *morphoHeart's* morphometric analysis suite can be utilised for high-quality and comprehensive tissue analysis, making it accessible for a wide range of experimental setups

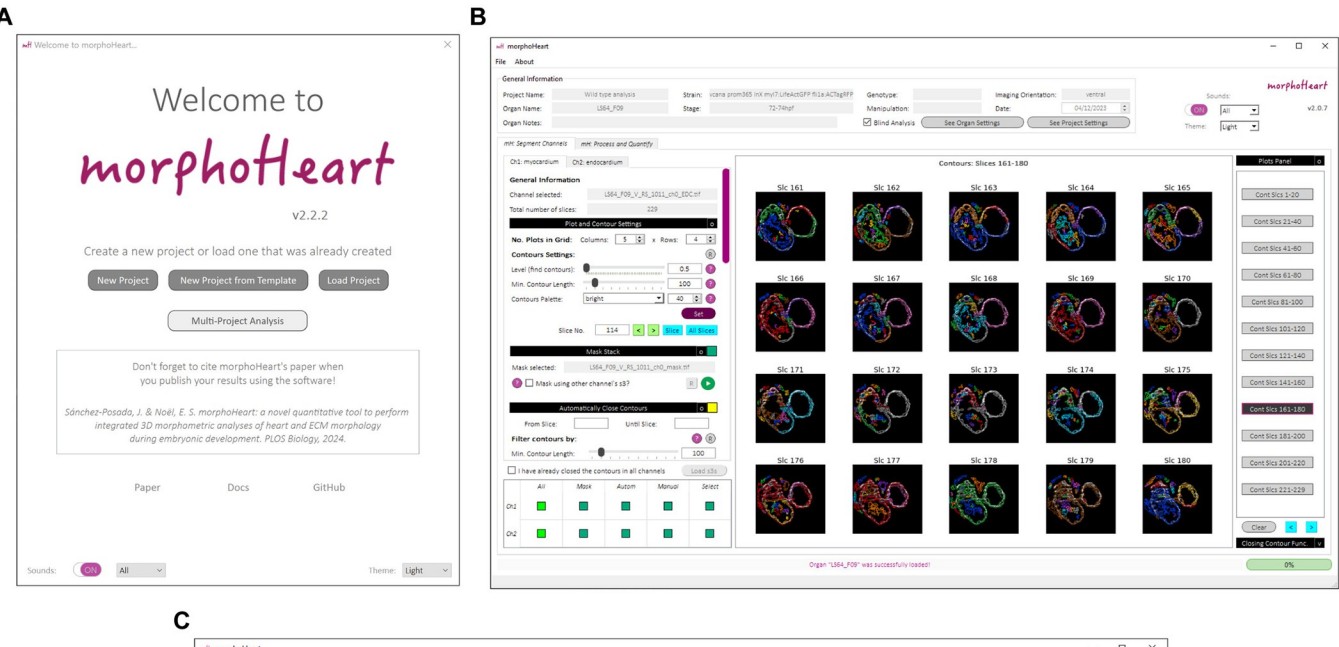

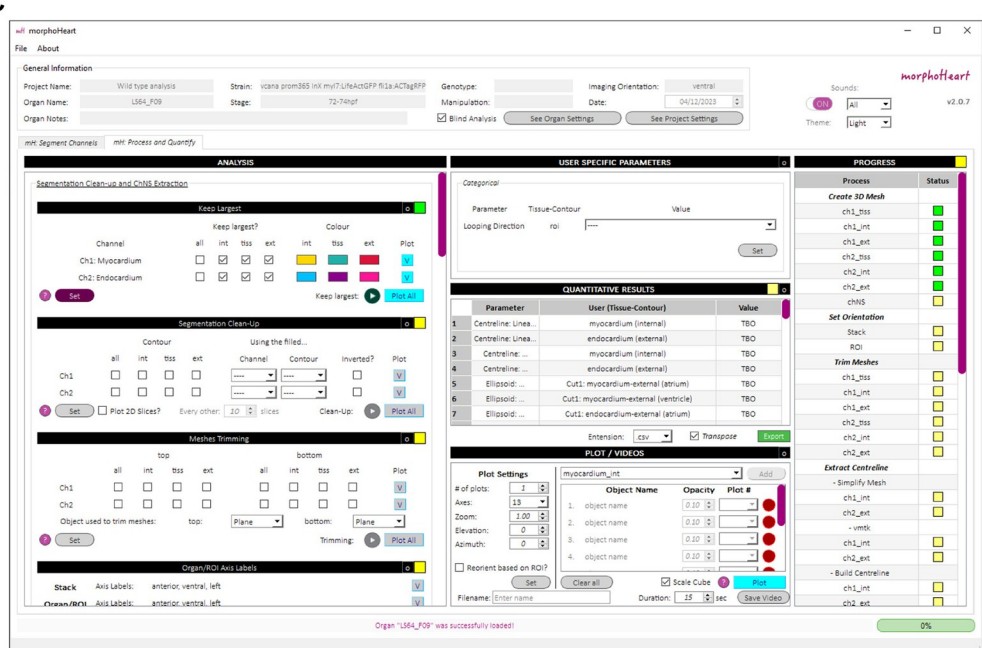

**Fig 8. *morphoHeart*'s GUI.** (A–C) Snapshots of morphoHeart''s GUI, showing the welcome window (A), segmentation tab (B), and the morphometric analysis tab (C).

and research needs. The design of this GUI expands *morphoHeart*'s use beyond cardiac tissue, enabling users to interact with the tools and access all its core functionalities seamlessly.

Looking forward, *morphoHeart* holds potential for a wide array of experimental applications. While it is not feasible to discuss every possible type of experiment, we envision several scenarios where *morphoHeart* can be particularly useful. For example, in the heart it could be employed to study trabeculation and ECM analysis across various developmental stages. Alternatively, it could be used to analyse other organs, in particular where the concept of "negative space" is relevant such as formation of the semi-circular canals in the developing ear.

Subcellular-level analyses may be possible, such as quantifying subcellular organelles within the myocardium, provided that the transgenic line can be segmented into clean contours. Although analysing individual organelles within each cell might be time-consuming, focusing on high magnification images of a cell or a group of cells would be more manageable. The *morphoCell* component of *morphoHeart* may also be applied to analysing distance between subcellular structures such as vesicles, since the coordinates of these structures could be input in a similar way to nuclei coordinates. In scenarios where precise measurements of proximity are required, such as the distance between adherens junctions and specific contours, *morphoHeart's* open-source release allows it to be modified by researchers with new modules to facilitate these analyses. These examples underscore the adaptability of *morphoHeart* and its potential to be customised by users to meet specific experimental needs.

Overall, *morphoHeart* represents a significant advancement in the field of morphometric analysis. Its comprehensive suite of tools, semi-automated segmentation capabilities, and ability to provide detailed 3D reconstructions make it an invaluable resource for studying organ morphology and tissue dynamics. As an open-source tool, *morphoHeart* is accessible to the broader research community, encouraging further development and customization to address a wide range of biological questions.

## Practical considerations for analysis of transgenic lines

To obtain the most accurate data using *morphoHeart*, the transgenic line and image acquisition approach need to be carefully evaluated and selected according to the biological question of interest. Some considerations to keep in mind are fluorophore selection and localization. As the cell morphology of each heart tissue is different (i.e., myocardial cells have uniform thickness and endocardial cells are thicker around the nucleus and thinner at junctions [68,72]), and the cellular arrangement within the tissue changes as the heart develops (i.e., monolayer, multilayer), some transgenic lines are better for studying heart morphology and tissue distribution using *morphoHeart*. Accordingly, to segment the endocardium, lines that label cell membranes are recommended to avoid "holes" in regions where the signal emission is very low due to thin cytoplasm. Bright, stable fluorophores are advised for deep heart imaging of the myocardium and long time-lapse studies to maintain image quality and prevent photobleaching and phototoxicity.

To inform these considerations, we performed a series of comparisons between transgenic lines, to understand their implications for data interpretation and experimental planning (S5 Fig). Importantly, these data highlight that absolute morphological parameters may vary depending on the transgenic line used. Due to the nature of these differences (particularly total chamber volume/elevated or reduced ballooning of the chamber), we speculate this represents subtle differences in baseline morphology or developmental timing due to different backgrounds of zebrafish lines, or insertional effects of individual transgenes. Our recommendations are therefore that in studies comparing organ dynamics over time, researchers are consistent in their use of transgenic lines at different stages to avoid introducing any line-specific confounding factors.

We also compared negative-space segmentation of the ECM to the *Tg(ubi:ssNcan-EGFP)* transgenic line, which acts as an HA-biosensor [6], at 48 to 50 hpf (S8 Fig), using *Tg(myl7: BFP-CAAX);Tg(ubi:ssNcan-EGFP)* embryos. We encountered difficulties in the automated thresholding of the *Tg(ubi:ssNcan-EGFP)* HA-sensor transgenic line, due to profound differences in intensity of the signal in different regions of the cardiac ECM, expression in the pericardium, and background in the pericardial cavity (S8A'–S8B' Fig). To overcome this issue and allow us to segment the ECM as cleanly as possible using this tool, we took the following

approach. We applied a threshold level to the contour-finding module in *morphoHeart* that allowed a relatively close segmentation of as much of the inner contour of the *Tg(ubi:ssNcan-EGFP)* signal as possible. This approach required manual contouring of a significant number of z-planes to ensure that the inner ECM contour in the ventricle was accurate and that parts of the inner ECM contour in the atrium were not missing. This approach does not allow accurate segmentation of the outer ECM contour, as the signal is often less bright, particularly in the atrium. Therefore, the "outer" ECM contour was defined as the inner myocardial contour using the *Tg(myl7:BFP-CAAX)* signal, and it was the combination of these 2 contours (inner myocardium and inner *Tg(ubi:ssNcan-EGFP)*) that were used to reconstruct the ECM (S8C Fig). We find in general that the ECM appears much thicker using this method than through negative-space segmentation (S8D–S8H Fig); however, the regionalisation of the ECM (thicker ECM in atrium compared to ventricle, thicker ECM on atrial outer curvature versus inner curvature) is still present (S8I and S8J Fig), demonstrating that negative-space segmentation analysis of ECM dynamics appears accurate. Our measurements of chamber-specific ECM volume using the *Tg(ubi:ssNcan-EGFP)* HA biosensor are in line with recently published values using manual segmentation of the same line [55]. There are a number of hypotheses for why we see these differences in total ECM volume using different methodologies: (1) challenges in contouring tightly to the *Tg(ubi:ssNcan-EGFP)* signal and/or the myocardium/endocardium signal used for negative-space segmentation, which may result in overestimation of ECM volume using the HA-sensor and/or underestimation using the negative-space segmentation approach; (2) the *Tg(ubi:ssNcan-EGFP)* transgenic line causes steric expansion of the ECM. The latter may be possible as the biosensor works by overexpressing a GFP tag which is bound to the HA-binding domain of Neurocan—meaning that the HA in the ECM is bound to extra components in the matrix which may affect ECM organisation or biophysical properties. We observe different ECM dynamics between 48 hpf and 72 hpf to those recently described [55], and it is possible that changes in native ECM composition, hydration, and crosslinking between these stages may be impacted by the presence of the HA biosensor. We therefore argue that label-free volumetric measurements, such as the approach used throughout this study, may be more accurate than non-native biosensors. While automated segmentation of the *Tg(ubi:ssNcan-EGFP)* is challenging across the whole heart, our data demonstrates that this line can be used to identify features of ECM distribution. We therefore recommend that for automated segmentation using this specific line, researchers may wish to restrict imaging and/or analysis to single chambers (countering the significant differences in signal intensity between chambers) and ideally image an adjacent tissue (e.g., myocardium to facilitate the delineation of the external contour of the ECM and isolate the background signal) to help obtain a more accurate ECM segmentation.

## Practical considerations for analysis of 2D heatmaps

The range of analytical outputs that *morphoHeart* generates allows for both visualisation of biological variation and conserved phenotypes in samples. For example, variation can be observed in quantitative parameters or individual organ heatmaps, while conserved features can be appreciated through averaged heatmap analysis. Variation in mutant phenotypes can be significant, and where such variation may exist we recommend that researchers visualise both 2D individual and averaged heatmaps to identify both well-conserved phenotypes, phenotypes that may be more variable in terms of expressivity, or phenotypes that are consistently observed but perhaps variable in location or positioning. Quantitative analyses for certain parameters (for example, volumetric measurements) should be used as an accessory method for visualising both overall phenotype and variability between samples.

While average heatmap analysis does require visual interpretation, the data generated can be used to quantitatively investigate and visualise variation. The 2D matrices that are used to construct each heatmap can be exported as.csv files, allowing the researcher to, for example, calculate the standard deviation for every position in the 2D matrix between different organs, providing a statistical measurement of variability across the tissue which could be represented as a 2D phenotype variability heatmap. These.csv files can also be "divided" into defined regions for more detailed analysis. Average values can be calculated for each region, enabling statistical comparisons (e.g., average myocardial ballooning in the outer versus the inner curvature of the atrium). This enables analysis of how measurements such as thickness or ballooning vary between regions, across developmental stages, or among different genotypes.

## Limitations

*morphoHeart* offers a novel suite of morphometric parameters for performing detailed quantifications and characterisations of heart morphogenesis. Some of the processes used to generate data (for example, quantifying tissue ballooning) requires the use of a centreline, which assumes a tubular nature to the structure that may not be appropriate or applicable in other tissue contexts. Thus, some functionality may remain limited to specific scenarios. However, the open-source nature of *morphoHeart* will allow other researchers to develop its analysis capabilities, implementing new quantifications or descriptors that support morphometric analysis of other tissues. A further acknowledged limitation of *morphoHeart* is the time required for segmentation, for example, cross-sectional analysis of ECM regionalisation such as that used in [7] is likely to be faster, but yields less accurate or comprehensive data with reduced parametric analysis. Therefore, the depth of analysis *morphoHeart* provides is unparalleled, but researchers may wish to factor time considerations into selecting it for analysis.

The processing and filtering steps used prior to *morphoHeart* segmentation were optimised for the myocardial and endothelial markers in the *Tg(myl7:lifeActGFP); Tg(fli1a:AC-TagRFP)* double transgenic line. These steps will likely require optimization dependent upon the transgenic line and image quality to allow accurate contour demarcation and tissue layer segmentation.

The 2D heatmaps generated in *morphoHeart* sometimes contain "gaps" corresponding to the inner curvature of the chambers. This is a function of mapping 3D heart geometry into a 2D matrix, and it is important to note that this tissue or signal is not missing per se, and we do not extrapolate or duplicate values into these "gaps" as this would constitute adding tissue that does not exist. Since the positioning of this gap is slightly different between hearts (as looping morphology is rarely identical), while the number of embryos that contributes to an averaged heatmap can be high, the number of points that contribute to the averaged inner curvature values may be less, as some hearts will have values to contribute and others will not. We therefore encourage researchers to always visualise both individual 3D and 2D heatmaps and compare 3D heatmaps to the 2D average heatmap to ensure a comprehensive understanding of the data. This practice allows for a better interpretation of the spatial relationships and variations within each sample, ensuring that important features are not overlooked. By analysing the individual contributions of each heart to the average data, researchers can gain insights into the variability present in their samples and assess how these gaps might affect their interpretations.

Finally, due to the approach used to obtain undisturbed 3D image data sets of the whole heart (temporary cessation of heartbeat), hearts were analysed with both chambers in a "relaxed" state, which is not directly representative of any stage in the cardiac cycle. Despite this limitation, the same approach was used for all the analysed embryos and so the

morphometric parameters obtained at the multiple developmental stages are comparable. The use of adaptive prospective optical gating [73] or macroscopic-phase stamping [74] imaging techniques in future studies will allow the acquisition of datasets at specific phases or throughout the whole cardiac cycle, which if combined with *morphoHeart's* capabilities could provide deeper understanding of the dynamic changes in tissue morphology during cardiac contraction.

The current version of *morphoHeart*, along with a detailed user manual, is available for download from GitHub https://github.com/jsanchez679/morphoHeart. An archived version of *morphoHeart* from the date of acceptance can be found at https://zenodo.org/records/14480354.

## Methods

### Resources

**Zebrafish husbandry.** Adult zebrafish were maintained according to standard laboratory conditions at 28.5°C. Zebrafish strains used are listed in Table 1. Embryos older than 24 hpf were treated with 0.2 mM 1-phenyl-2-thiourea (PTU) in E3 medium to inhibit melanin production. All animals were euthanized by immersion in overdose of Tricaine (1.33 g/l). Animal work was approved by the local Animal Welfare and Ethical Review Body (AWERB) at the University of Sheffield, conducted in accordance with UK Home Office Regulations under PPLs PA1C7120E and PP4569114, and in line with the guidelines from Directive 2010/63/EU of the European Parliament on the protection of animals used for scientific purposes.

### Generating the *myl7:Citrine* transgenic zebrafish line

The coding sequence of Citrine was amplified from a template using the following primers: F: 5′ -ggggacaagtttgtacaaaaaagcaggcttc<u>GCCGCCACC</u>**ATG**GTGAGCAAGGGCGAGGAGCTGT-3′; R: 5′ -ggggaccactttgtacaagaaagctgggtt**TTA**CTTGTACAGCTCGTCCATGCCG-3′. Start and stop codons are highlighted in bold, Kozak sequence underlined, attB sites in lower case. The Citrine PCR product was ligated into pDONR221 [80] using standard tol2kit protocols to

**Table 1. Zebrafish strains and study resources.** Zebrafish strains, software resources, and general chemical reagents used in this study.

| REAGENT OR RESOURCE | SOURCE | IDENTIFIER |
|---|---|---|
| **Experimental models (organisms/strains)** | | |
| *Tg(myl7:lifeActGFP)* | See Reischauer and colleagues [75] | s974Tg |
| *Tg(myl7:BFP-CAAX)* | See Guerra and colleagues [76] | bns193Tg |
| *Tg(ubi:ssNcan-EGFP)* | See Grassini and colleagues [6] | uq25bhTg |
| *Tg(myl7:H2B-mScarlet)* | See Boezio and colleagues [77] | bns534Tg |
| *Tg(fli1:Actin-CB-TAGRFP)* | See Savage and colleagues [78] | sh511Tg |
| *hapln1a*^sh580 | See Derrick and colleagues [7] | sh580 |
| *Tg(myl7:Citrine)* | This study | sh642Tg |
| **Software and algorithms** | | |
| Vision4D | arivis | https://www.arivis.com/ |
| *morphoHeart* | This study | https://github.com/jsanchez679/morphoHeart |
| FIJI | See Schindelin and colleagues [79] | https://fiji.sc/ |
| Pre-*morphoHeart* Masking and Cropping FIJI Macro | This study | See S1 Data |
| **Chemicals, peptides, and recombinant proteins** | | |
| Ethyl 3-aminobenzoate methanesulfonate salt (Tricaine) | Sigma | A5040 |
| 1-phenyl2-thiourea (PTU) | Sigma | P7629 |

generate pME-Citrine. p5E-myl7promoter [81], pME-Citrine and p3E-poly(A) [80] were ligated into pDestTol2pA3 [80] using standard tol2kit protocols; 50 pg of the *myl7:Citrine-poly (A)* plasmid was co-injected with 50 pg of *tol2* transposase [80] mRNA in a 1 nL volume into the cell of 1-cell stage WT (AB) embryos, and fluorescent embryos were grown to adulthood. F0 adults were outcrossed to WT to identify founders with suitable germline transmission. The line was established from F1 individual with a single transgenic insert transmitting at 50% to F2s.

## Lightsheet imaging

To assess cardiac morphology at different developmental stages, live or fixed zebrafish embryos were imaged on a ZEISS Lightsheet Z.1 microscope. To stop the heartbeat of live embryos and aid image analysis, prior to mounting, 3 to 5 embryos were anesthetised by transferring them from a dish containing E3+PTU to a new cooled dish containing E3 and 8.4% Tricaine (E3+Tricaine). Anesthetised embryos were embedded in 1% low melting point agarose with 8.4% of Tricaine in black capillaries (1 mm diameter; Brand 701904). To ensure the heartbeat was arrested during the acquisition, the imaging chamber was filled with E3+Tricaine and maintained throughout the experiment at 10˚C.

All images were acquired using a 20× objective lens with 1.0 zoom. Single-side lasers with activated pivot scan were used for sample illumination. High-resolution images capturing the whole heart were obtained with 16 bit image depth, $1,200 \times 1,200$ pixel ($0.228$ μm $\times$ $0.228$ μm pixel size resolution) image size and $0.469$ to $0.7$ μm $z$-stack interval. For double fluorescent transgenic embryos, each fluorophore was detected on separate channels.

## Image preprocessing

To remove noise artefacts, accentuate details, and enhance tissue borders, raw lightsheet *z*-stacks for each tissue channel were processed and filtered in arivis Vision4D. To smooth noisy regions but preserve the edges of each tissue layer, the *Denoising Filter (3D)* was applied to the RAW data set. The resulting images were then processed using the *Background Correction* filter to reduce variations in intensity throughout the whole image set. Next, the *Morphology Filter* was used to sharpen the tissue borders, followed by *Membrane Enhancement* to boost the signal of membranes, producing clear slices with enhanced and sharpened borders in each channel.

To further enhance signal and reduce file size, after individual channels were processed in Vison4D, images were then processed in Fiji. First any residual salt-and-pepper noise was removed using the *Despeckle* filter. An *Enhancement* filter was then applied to both channels to improve the contrast of the images without distorting the grey level intensities. Finally, a Maximum Intensity Projection (MIP) of a composite containing both processed channels was used to define a square that contains the region of interest (ROI) comprising the heart. This ROI was used to crop each channel reducing the image size to be imported into *morphoHeart* for segmentation. Image J Macro is in S1 File.

## Statistical analysis

All data groups were initially analysed for normality using the Shapiro–Wilk Normality test. In data sets containing 2 groups (e.g., wild-types and homozygous mutants) comparative statistics of normally distributed data sets were carried out using Un-paired *t* test, while non-normal data sets were analysed using Mann–Whitney test. In data sets containing more than 2 groups, comparative statistics of normally distributed data sets were carried out using one-way

ANOVA (post hoc: Tukey's Multiple Comparisons), while non-normal data was analysed using Kruskal–Wallis (post hoc: Dunn's Multiple Comparisons).

## Supporting information

**S1 Fig. Image acquisition.** (A) Schematic overview of *morphoHeart*'s pipeline. (B–D) Overview of the sample acquisition and preparation for the data presented in this manuscript. The hearts of zebrafish larvae were temporarily arrested in a lightsheet chamber (B), and *z*-stack images acquired (C) which were preprocessed prior to use in *morphoHeart* (D). E: Example slices at progressive z-planes through a processed and filtered z-stack of a 72hpf *Tg(myl7:life-ACT:GFP);Tg(fli1a:AC-TagRFP)* heart, with the myocardial actin labelled in green and endothelial actin labelled in magenta. (F) Snapshots of 3D renderings of processed and filtered 72–74 hpf hearts using ImageJ's 3D Viewer from a ventral, left, and dorsal view. (G) Snapshots of the 3D segmented heart (same hearts as shown in F) using *morphoHeart* from a ventral, left, and dorsal view.
(TIF)

**S2 Fig. Acquisition of tissue "centrelines" and segment orientation.** (A) Schematics depicting both linear (green) and looped (blue) lines through the heart between inflow and outflow poles. The blue looped line represents the "centreline" through the tissue generated by assuming the heart as a tubular structure. (B) Internal Voronoi diagram generated when extracting the centreline using the vmtk package integrated into morphoHeart. (C) Internal myocardial mesh filled with maximum-inscribed spheres used to calculate each point of the centreline. (D) Myocardial tissue meshes showing linear heart length (green, linear distance between poles) and looped heart length (blue, vmtk-calculated centreline through tissue between poles). (E–N) Heart morphogenesis is accompanied by relative chamber realignments. Screenshot depicting *morphoHeart*'s method for measuring chamber orientations from the heart's ventral face (E), and schematic depicting how each chamber angle is calculated with respect to a reference vector (F). The atrium remains static (G), while the ventricle first moves substantially clockwise, and after looping pivots counter-clockwise (H), resulting in a frontal displacement and realignment of the chambers as the heart loops, grows, and compacts (I). Screenshot depicting *morphoHeart*'s method for measuring lateral chamber orientations (J), and schematic depicting how each chamber lateral angle is calculated with respect to a reference vector (K). Each chamber angle is measured, and the lateral angle between them calculated (N). The atrium's lateral position remains relatively unchanged (L), while the ventricle straightens as the heart loops, and becomes laterally displaced as it compacts (M). This results in repositioning of the chambers or lateral rotation around the AVC (N). One-way ANOVA with multiple comparisons. * $p < 0.05$, ** $p < 0.01$, *** $p < 0.001$, **** $p < 0.0001$; 34–36 hpf: $n = 9$; 48–50 hpf: $n = 10$; 58–60 hpf: $n = 8$; 72–74 hpf: $n = 10$. Plots display median and quartiles. The numerical data underlying this figure can be found in S1 Data.
(TIF)

**S3 Fig. The chambers undergo different geometrical shape changes as the heart develops.** (A) Illustration of the geometrical measurements acquired for both atrium (A) and ventricle (V), including chamber depth (d), length (L) and width (w). (B) Individual chamber meshes (magenta) are projected to the anterior (yellow), right (red), and dorsal (blue) face of a reference organ cube, and its largest dimensions are used to create an ellipsoid that best resembles its shape (green). (C–H) Quantification of chamber depth (C, F), length (D, G), and width (E, H). The atrium expands in the z-plane, becoming deeper throughout development (C). Between 48 hpf and 74 hpf it also shortens (D) and narrows (E), becoming more spherical.

The ventricle also expands in depth (F), while it lengthens (G) between 34 hpf and 50 hpf to become a more elongated shape. One-way ANOVA with multiple comparisons. * $p < 0.5$, ** $p < 0.01$, *** $p < 0.001$, **** $p < 0.0001$; 34–36 hpf: $n = 9$; 48–50 hpf: $n = 10$; 58–60 hpf: $n = 8$; 72–74 hpf: $n = 10$. Plots display median and quartiles. The numerical data underlying this figure can be found in S1 Data.
(TIF)

**S4 Fig. Generation of individual and averaged 2D heatmaps.** (A) Schematic depicting the method for unrolling 3D heatmaps into 2D heatmaps. The centreline (green) is extended beyond the poles of the heart, and a defined number of planes are cut through the heart mesh, transverse to the orientation of the centreline at each plane. For each intersecting plane, the ventral-most point of the mesh is cut and assigned position 0˚, while the dorsal-most point of the mesh is cut and assigned position 180˚/−180˚, giving each mesh point in each plane a universal coordinate. As each mesh point is additionally associated to a measured thickness/ballooning value, together this allows the tube to be "unrolled" and mapped onto a standard 2D geometry (B). (C, D) As each heatmap has the same coordinate system, the thickness/ballooning measurement of multiple organs (at the same stage and of the same genotype; C) can be averaged at each coordinate, allowing individual heatmaps from multiple embryos to be combined, producing an average heatmap which represents typical phenotype independent of small biological variation in tissue morphology (D). (E, F) Schematics to facilitate interpretation of the heatmaps, demonstrating correspondence of the regions identified in the 2D heatmaps (E) and the 3D morphology of the organ (F), identified with different colours for each chamber. Labels around the heatmap indicate cardiac region: D—dorsal, V—ventral, L—left, R—right, AOC—atrial outer curvature, AIC—atrial inner curvature, VOC—ventricular outer curvature, VIC—ventricular inner curvature, AV Canal—atrioventricular canal.
(TIF)

**S5 Fig. Comparison of different myocardial transgenic lines.** (A, B) 3D heatmap analysis of chamber (myocardial) ballooning in *Tg(myl7:lifeActGFP)* transgenic hearts (A) and *Tg(myl7: BFP-CAAX)* transgenic hearts (B) at 48–50 hpf. (C, D) 3D heatmap analysis of myocardial thickness in *Tg(myl7:lifeActGFP)* transgenic hearts (C) and *Tg(myl7:BFP-CAAX)* transgenic hearts (D) at 48–50 hpf. Parts of the dorsal atrial myocardium are missing in *Tg(myl7: BFP-CAAX)* transgenic embryos (yellow arrow, D). (E) Quantitative analysis of chamber volume reveals that ventricular chamber volume is smaller in *Tg(myl7:BFP-CAAX)* hearts compared to *Tg(myl7:lifeAct-GFP)* transgenics. (F) Quantitative analysis of myocardial volume reveals that atrial myocardium volume is smaller in *Tg(myl7:BFP-CAAX)* hearts compared to *Tg(myl7:lifeAct-GFP)* transgenics. (G) Analysis of looping ratio revealing that looping geometry is not significantly different between transgenes. (H, I) 3D heatmap analysis of chamber (myocardial) ballooning in *Tg(myl7:lifeActGFP)* transgenic hearts (H) and *Tg(myl7:Citrine)* transgenic hearts (I) at 72–74 hpf. (J) Quantitative analysis of chamber volume reveals that atrial volume is larger and ventricular chamber volume smaller in *Tg(myl7:Citrine)* hearts compared to *Tg(myl7:lifeAct-GFP)* transgenics. (K) Quantitative analysis of myocardial volume reveals that atrial myocardium is increased in *Tg(myl7:Citrine)* hearts compared to *Tg (myl7:lifeAct-GFP)* transgenics. (L) Analysis of looping ratio revealing that looping geometry is not significantly different between transgenes. One-way ANOVA with multiple comparisons. * p < 0.5, ** p < 0.01, *** p < 0.001, **** p < 0.0001. *Tg(myl7:BFP-CAAX)* 48 hpf n = 5; *Tg (myl7:lifeActGFP)* 48 hpf n = 10; *Tg(myl7:Citrine)* 72 hpf n = 5; *Tg(myl7:lifeActGFP)* n = 10. Plots display median and quartiles. The numerical data underlying this figure can be found in S1 Data.
(TIF)

**S6 Fig. Comparison of fixed vs. live heart morphology.** (A, B) 3D heatmap analysis of chamber (myocardial) ballooning in live (A) and fixed (B) *Tg(myl7:lifeActGFP)* transgenic embryos at 48–50 hpf. Arrow in (B) depicts folding of the atrial tissue in fixed samples that is not observed in live hearts. (C, D) Averaged 2D unrolled chamber ballooning heatmaps in live (C) and fixed (D) embryos, highlighting the reduction in chamber ballooning in fixed tissue. (E) Quantitative analysis of chamber volume reveals that both chambers in fixed hearts are smaller than live counterparts. (F) Quantitative analysis of myocardial volume demonstrates that tissue shrinkage occurs in fixed samples. (G) Analysis of looping ratio revealing that looping geometry is not significantly affected by fixation. One-way ANOVA with multiple comparisons. *** $p < 0.001$. *Tg(myl7:lifeActGFP)* 48 hpf live $n = 10$; *Tg(myl7:lifeActGFP)* 48 hpf fixed $n = 5$. Plots display median and quartiles. The numerical data underlying this figure can be found in S1 Data.
(TIF)

**S7 Fig. *morphoCell* module facilitates analysis of cell number and size.** (A) Flow diagram describing the phases involved in the process of acquiring comprehensive cell data using *morphoCell*. Nuclei identification is performed using Imaris spot-finder function and given as input into *morphoCell* with the original z-stack images. (B) Maximum intensity projection of the heart of a *Tg(myl7:BFP-CAAX); Tg(myl7:H2B-mScarlet)* transgenic embryo, highlighting the myocardium (blue) and myocardial nuclei (magenta). (C–E) *morphoCell* renderings of myocardium and cardiomyocyte nuclei (spheres). Nuclei are initially not categorised into chambers; however, chambers can be separated via a user-defined plane, and nuclei subsequently automatically categorised as atrial (C; orange spheres) or ventricular (C; green spheres). (D–F) Cells identified in each chamber (atrium, ventricle) are clustered into distinct groups containing "seed" cells (single colour) or "neighbouring" cells (striped) that are representative of cell organisation within the chamber (D: atrial cluster, comprised of a central seed cell and 3 neighbouring cells, E: ventricular clusters, comprised of a central seed cell and 4 neighbouring cells). Seeds and neighbouring cells can be categorised to regions of interest within the chamber. The 3D distances between the seed cell and its neighbouring cells are measured and averaged to generate a single average internuclear distance value for each seed (F).
(TIF)

**S8 Fig. Comparison of ECM segmentation methods.** (A–B') Example z-slices through *Tg (myl7:BFP-CAAX)* and *Tg(ubi:ssNcan-EGFP)* double transgenic hearts, highlighting the myocardium (A, B) and HA-biosensor (A', B'), including contours automatically detected by morphoHeart. The myocardial signal shows tight and accurate contouring on both the outer and inner surface of the tissue, whereas the HA-biosensor signal around the heart is very variable, with variability in how tightly the internal ECM contour fits to the signal in the ventricle (white arrow), and failure to identify the outer contour of the cardiac ECM in the ventricle (yellow arrow). (C) 3D reconstructions of the internal myocardial mesh reconstruction (yellow), inner HA-biosensor mesh reconstruction (blue), and the resulting ECM mesh reconstruction (orange) generated by combining the first 2 meshes. (D, E) 3D heatmap reconstructions of cardiac ECM thickness in the negative-space ECM segmentation hearts (D) and the *Tg(ubi:ssNcan-EGFP)* hearts (E). (F, G) Average 2D unrolled ECM thickness heatmaps of negative-space ECM segmentation hearts (F) and *Tg(ubi:ssNcan-EGFP)* hearts (G). The cardiac ECM in *Tg(ubi:ssNcan-EGFP)* transgenic hearts appears generally thicker than that derived from negative-space segmentation, but shows regional distribution. (H–J) Comparative quantitative analysis between *Tg(ubi:ssNcan-EGFP)*-derived ECM and negative-space-derived ECM volumes in whole hearts (H), atria (I), and ventricles (J). In general, ECM

volume is increased in the *Tg(ubi:ssNcan-EGFP)* HA-biosensor line compared to negative space segmentation, but the same patterns of ECM regionalisation (higher ECM volume in atrium compared to ventricle, higher ECM volume on atrial outer curvature) are observed. One-way ANOVA with multiple comparisons. ** $p < 0.01$, *** $p < 0.001$. *Tg(ubi:ssNcan-GFP)* 48 hpf $n = 5$; negative-space segmentation 48 hpf $n = 10$. Plots display median and quartiles. The numerical data underlying this figure can be found in S1 Data.
(TIF)

**S9 Fig. Relative ECM contribution to total heart volume is disrupted in *hapln1a* mutants.** (A–C) Analysis of heart composition by percentage contribution of each compartment of the heart. Schematic depicts contributing tissues/regions of the heart (A). During early stages of heart morphogenesis, hapln1a mutants display a comparative reduction in contribution of ECM to total heart or atrial volume (B, C). (D–G) Averaged 2D ECM thickness heatmap reveals that while the magnitude of ECM expansion and regionalisation is reduced in hapln1a mutants at 34–36 hpf (F) and 48–50 hpf (G), a small region of the atrium still exhibits a thicker ECM. Labels around the heatmap indicate cardiac region: D—dorsal, V—ventral, L—left, R—right, AOC—atrial outer curvature, AIC—atrial inner curvature, VOC—ventricular outer curvature, VIC—ventricular inner curvature, AV Canal—atrioventricular canal. Plots display mean values for each category. The numerical data underlying this figure can be found in S1 Data.
(TIF)

**S1 File. Pre-morphoHeart Masking and Cropping FIJI Macro.**
(DOCX)

**S1 Data. Excel spreadsheet containing, in separate sheets, the underlying numerical data for Figs 2J, 2K, 2L, 2N, 2O, 2P, 2Q, 3B, 3C, 4B, 4C, 4D, 4E, 4G, 4H, 5A, 5C, 5D, 6C, 6D, 6Q, 6R, 7C, 7D, 7E, 7H, 7I, 7J, 7K, 7L, 7M, 7N, S2G, S2H, S2I, S2L, S2M, S2N, S3C, S3D, S3E, S3F, S3G, S3H, S5E, S5F, S5G, S5J, S5K, S5L, S6E, S6G, S8H, S8I, S8J, S9B and S9C.**
(XLSX)

# Acknowledgments

We are grateful to Eric Pollitt, Emma Armitage, Chris Chan Jin Jie, Yangsheng Zhou, Enze Wang, and Angelica Spadaro for testing early versions of *morphoHeart*. The plasmid containing the Citrine target sequence was a kind gift from Henry Roehl. We thank Eric Pollitt and Tanya Whitfield for critical reading of the manuscript. Lightsheet imaging was performed at the Wolfson Light Microscopy Facility.

# Author Contributions

**Conceptualization:** Juliana Sánchez-Posada, Emily S. Noël.

**Data curation:** Juliana Sánchez-Posada, Emily S. Noël.

**Formal analysis:** Juliana Sánchez-Posada, Emily S. Noël.

**Funding acquisition:** Emily S. Noël.

**Investigation:** Juliana Sánchez-Posada, Emily S. Noël.

**Methodology:** Juliana Sánchez-Posada, Emily S. Noël.

**Project administration:** Emily S. Noël.

**Resources:** Juliana Sánchez-Posada, Christopher J. Derrick, Emily S. Noël.

**Software:** Juliana Sánchez-Posada.

**Supervision:** Emily S. Noël.

**Visualization:** Juliana Sánchez-Posada, Emily S. Noël.

**Writing – original draft:** Juliana Sánchez-Posada, Emily S. Noël.

**Writing – review & editing:** Juliana Sánchez-Posada, Christopher J. Derrick, Emily S. Noël.

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
