## [Editor Report · Decision Letter 0]

21 Mar 2024

Dear Dr Noël, 

Thank you for submitting your manuscript entitled "morphoHeart: a novel quantitative tool to perform integrated 3D morphometric analyses of heart and ECM morphology during embryonic development" for consideration as a Methods and Resources Article by PLOS Biology. Please accept my apologies for the delay in getting back to you as we consulted with an academic editor about your submission. 

Your manuscript has now been evaluated by the PLOS Biology editorial staff, as well as by an academic editor with relevant expertise, and I am writing to let you know that we would like to send your submission out for external peer review.

Once your full submission is complete, your paper will undergo a series of checks in preparation for peer review. After your manuscript has passed the checks it will be sent out for review. To provide the metadata for your submission, please Login to Editorial Manager (https://www.editorialmanager.com/pbiology) within two working days, i.e. by Mar 23 2024 11:59PM.

Kind regards,

Richard

Richard Hodge, PhD

rhodge@plos.org

PLOS

---

## [Decision Letter · Decision Letter 1]

21 May 2024

Dear Dr Noël,

Thank you for your patience while your manuscript "morphoHeart: a novel quantitative tool to perform integrated 3D morphometric analyses of heart and ECM morphology during embryonic development" was peer-reviewed at PLOS Biology. It has now been evaluated by the PLOS Biology editors, an Academic Editor with relevant expertise, and by three independent reviewers. 

In light of the reviews, which you will find at the end of this email, we would like to invite you to revise the work to thoroughly address the reviewers' reports.

As you will see below, the reviewers are all positive, and think that morphoHeart will be a useful tool with wide applications. However, they also have some concerns that should be addressed before we can consider the manuscript for publication. Particularly, we would need you to show analyses on a different combination of transgenic animals to assess the consistency of the method, as suggested by Reviewer 2. You should also include all the comparisons suggested by Reviewer 1, along with clarifications on the software, among other issues.

Given the extent of revision needed, we cannot make a decision about publication until we have seen the revised manuscript and your response to the reviewers' comments. Your revised manuscript is likely to be sent for further evaluation by all or a subset of the reviewers.

**IMPORTANT - SUBMITTING YOUR REVISION**

*Re-submission Checklist*

*Published Peer Review*

*PLOS Data Policy*

*Blot and Gel Data Policy*

Sincerely,

Suzanne

Suzanne de Bruijn, PhD

Associate Editor

PLOS Biology

sbruijn@plos.org

on behalf of

Richard Hodge, PhD

Senior Editor

PLOS Biology

Reviewers' comments

Reviewer #1: Morphogenesis is an incredibly complex and dynamic process. The ability to accurately and comprehensively report morphological findings has been hampered by the dearth of open-source, 3D tools that are navigable for investigators without extensive programming ability. With a rich history in cardiac developmental biology, Sáchez-Posada and Noël were well-positioned to understand the needs of the field, and, in this manuscript, the authors present a newly developed program, morphoHeart, that will aid researchers in understanding both wild-type morphology and specific deviations from it in various contexts. With the extensive resources associated with this manuscript and what appears to be a very user-friendly interface, this reviewer foresees the potential for wide application of morphoHeart. Minor additions, clarifications, and explicit instructions for the potential user may increase the uptake of morphoHeart in the developmental biology community, and beyond. Indeed, the product itself appears quite complete in its usefulness; with this in mind, this reviewer supplies suggestions for manuscript revision that may boost the uptake of this software in the community.

To help potential users determine whether morphoHeart is right for them:

1. Although the authors provide well-reasoned arguments for using live tissue as input for their particular experimental questions, there will certainly be scenarios where researchers need or want to use fixed tissue. It would be helpful for these readers to see a brief comparison between fixed and lived tissue, even if this is a simple comparison, such as some volume comparisons at one developmental stage. If such data aren't able to be included, it would be helpful to include at least a section in the Discussion that would inform users of anything to keep in mind if trying to use fixed tissue.

2. The ability to quantify "negative space" is indeed an interesting and attractive aspect of this program. However, others interested in the ECM itself may benefit from a brief discussion of the usage of actual ECM markers as input for morphoHeart. Have the authors tried this approach and compared it to the quantification of empty space? Are there certain scenarios where one might choose the ECM marker approach over the empty space approach?

3. The reviewer appreciates the authors' morphoHeart analysis of the hapln1a mutant as a comparison to previous methods. However, it seems this comparison could be even more useful with a few additions: 

a. It would be interesting to see what the hapln1a ECM thickness heatmap would look like at 34-36 hpf, particularly given the early ECM phenotype shown in Derrick et al. (2022) and the difference in atrial ECM volume at 34-36 hpf in Fig. 7K.

b. It seems that morphoHeart will be particularly useful as a "phenotype finder" for a new or relatively uncharacterized mutant, and one can imagine that the regional expansion heatmaps (as in Fig. 3) would be especially informative to help one hone in on a specific region of an organ for further study. The authors use several of the tests in their arsenal to explain the dysmorphia of the hapln1a heart, but it seems a missed opportunity not to show a comparison to wild-type regional expansion at 50 hpf. 

c. Given the parallels that the authors can now make between their current and previous analyses of hapln1a mutants, a discussion of the pros and cons of these approaches would be satisfying. For example, what can this new method tell us about the nature of the ECM in hapln1a mutants that couldn't be seen from the types of analyses in Derrick et al. (2022)? Ultimately, in what scenarios do the authors suggest that investigators choose one approach over the other — or is there room and necessity for both?

4. Of course, it's not reasonable for the authors to discuss every possible type of experiment for which users might employ morphoHeart/morphoCell in the future. However, it may be worthwhile for the authors to provide their vision for the categories of experiments that are possible now or with some modification of the software, and what categories are likely out of bounds. For example, could they comment on the feasibility of quantifying the percentage of myocardial volume occupied by Golgi, or the proximity of adherens junctions to a particular contour?

Clarifications on software methodology:

1. Line 127: The sentence "To extract the myocardium and endocardium, individual slices making up each channel go through a process of contour detection and selection, extracting the contours that delineate the tissue layer." implies that contour identification is automatic, but explicit inclusion of this fact would be edifying to potential users. Additionally, this may be a good place to include some instruction on what makes a "good" marker for morphoHeart (perhaps a briefer version of the "Transgenic Line Selection" from the User Manual).

2. Readers may wonder about the projection method used to unroll the heatmaps, particularly because there seems to be missing signal in the VIC in Fig. 3M-O, for example. The reviewer suggests a brief discussion of the process and the potential for associated issues when introducing the first usage of this projection method.

3. It is somewhat unclear at what point and in which situations samples are "averaged" into a stereotypical organ. A brief discussion on this point should help readers to understand the types of analyses for which they can expect to be able to visualize biological variation (e.g. ECM volume), and when this is not possible (e.g. ECM heatmaps). Along these lines, what would be best practices for analysis of a mutant phenotype that includes a lot of morphological variation between individuals (so as to avoid missing key elements of the phenotype)?

Additional instructions for usage:

1. What are the exact system requirements for the software to run? Currently, the User Manual states "morphoHeart currently works only in Windows", implying a fairly rudimentary system is acceptable. However, an explicit statement of this would make the software especially attractive to potential users (or in contrast, if a more sophisticated system is necessary, this information is especially important). Because this may change from version to version, and on different platforms (for example, when morphoHeart becomes available for other operating systems), the User Manual is a perfectly acceptable place for this information.

2. Some more guidance from the authors on experimental setup may make potential users more comfortable. For example: Are other imaging setups acceptable (i.e. confocal instead of lightsheet)? What is the optimal step size between slices? Is there an upper limit to number of slices (or total file size) imported into the software?

Reviewer #2: Summary

The manuscript by Possada and Noel presents morphoHeart, a new computational tool that allows for quantitative analysis of heart morphology in 3D. Using zebrafish as a model, the authors use this newly developed software to track heart development between 34 and 74 hours post fertilization (hpf). morphoHeart extracts external and internal contours of the two different layers of the heart, i.e. the myocardium (labeled with Tg(myl7:lifeact-GFP) and the endocardium (Tg(fli1a:TagRFP)), allowing for volumetric and geometric quantification of the cardiac tissue and the lumen. It can calculate the centerline of the heart, which allows for assessment of angular evolution during the looping process (looping ratio) and of tissue thickness maps in 2D. The authors quantified the extracellular space/ECM between the two tissues by quantifying the negative space between them. Through morphoHeart, the authors can distinguish regional differences in the heart at the tissue and the ECM levels, not only between atrium and ventricle, but also between inner and outer curvature of the heart. 

Overall, the authors presented a very nice tool that would be useful for scientists working in the field of morphogenesis. The analysis appears strongly quantified and detailed, with regional distinctions that have not been clearly reported in previous publications. The software also appears to be generalizable to many different 3D systems with well-established fluorescent tools/markers. Currently, the manuscript mostly focuses on descriptive features of the developing heart, except for differences between wild types and hapln1a mutants. However, future work from the authors and others in the field would benefit from this software in analyzing different genetic conditions and loss-of-function phenotypes. 

As I am not an expert in computational development, I will only comment on the biological findings of this study and how the quantitative analysis might be improved. 

Major comments

- The sample sizes are not very big (in most cases, n is 10 or lower), which might account for the high variability of the values particularly in 72-74 hpf animals. The data might also not necessarily have a normal distribution due to the limited sample size. It would be better evidence if the authors could expand their numbers. 

In particular, the authors could use a different combination of transgenic animals than the one already used in the paper (Tg(myl7:lifeact-GFP), Tg(fli1a:TagRFP)), e.g. with a red myocardial marker and a green endocardial marker, to assess the consistency of their results. This would also alleviate any concerns of non-specific effects, in particular because lifeact-GFP could exert side effects for cell morphology and development, and confirm the specificity of their results. The authors have also already performed imaging with Tg(myl7:BFP-CAAX); were the results of myocardial volume measurements consistent with what was observed with Tg(myl7:lifeact-GFP)?

- Depending on the angle of the embryo mounting and depth of tissue, fluorescence signal intensity can be difficult to detect throughout the whole heart. Have the authors accounted for thorough detection to account for the entire volume of the heart or the lumen, which might result in detection of atrial volume reduction at 74 hpf? Including representative images of the z-stack or a 3D reconstruction of the heart in Supplemental figures would be instructive. 

- Considering that morphoCell was performed on "Tg(myl7:BFP-CAAX); Tg(myl7:H2B-mScarlet", did the authors assess whether the individual sizes/volumes of CMs can be measured using the membrane marker instead of the inter-nuclear distance or myocardial wall thickness? If not, what are the limitations of the software that prevent the use of the BFP signal? 

- The finding that the ventricle ECM significantly expands only in a very limited time window (between 34 and 48 hpf) and immediately decreases at 56 hpf is surprising. Mechanistically, how do the authors speculate this occurs (purely by degradation?) and what would be the functional consequence were this expansion to be abolished? Do the authors have a model in which this ECM expansion between 34 and 48 hpf is abolished?

- Similarly, what do the authors speculate is the reason behind the reduction of atrial myocardial tissue volume after 48 hpf? 

- The ECM regionalization appears to be primarily driven by cardiac trabeculation. What happens to the ECM volumes in the different regions in animals with no trabeculae? It would be interesting to test out morphoHeart in ErbB2-inhibited animals. 

Minor comments

- Specify what "GUI" stands for in the abstract. 

- I would use a term other than "degradation", particularly when describing the hapln1a mutant phenotype, as there is no direct evidence yet that the ECM is actively being removed. 

- On page 14, "We reveal a novel requirement for in driving expansion of the atrial lumen", the word "Hapln1a" is missing. 

- The abbreviation "SHF" needs to be clarified. 

- Scale bars are missing from Figure 6F and 4G.

- Is 10^5 correct on the y axis of the graphs in Figure 7H-J?

- Figure Legend 3H-J should be H-K. 

Reviewer #3: The manuscript by Sánchez-Posada and Noël aims to fill a gap in analysing early heart development, particularly heart morphogenesis in live embryos. Existing methods have limitations, like collapse or shrinkage of tissue due to fixed tissue, which can alter results. The authors propose morphoHeart as a tool to fill the current gap by offering a comprehensive pipeline for semi-automated segmentation and analysis of developing hearts, including segmentation of the cardiac ECM. The software's graphical user interface expands its utility beyond cardiac tissue, allowing analysis of multiple tissue layers and, importantly, negative space segmentation. The latter is one of the significant contributions of morphoHeart - its ability to provide negative space segmentation of the cardiac ECM, a feature not available in other analytic approaches. Additionally, it provides high resolution and methodical detail in quantifying cardiac tissues of developing zebrafish hearts. The open-source nature of morphoHeart is a critical aspect of its contribution. It enables other researchers to expand its capabilities, implementing new quantifications or descriptors to support analysis of various tissues, making it attractive also to non-cardiac users. This open-access approach fosters collaboration and innovation in the field; therefore the authors should be commended by their effort.

The paper presents a detailed analysis of the zebrafish heart and uses morphoHeart to systematically characterise in 3D how the heart undergoes periods of growth and compaction during early morphogenesis, including quantifying distinct geometric changes during heart looping and chamber expansion, and the regionalised and chamber-specific ECM volumetric remodelling that happens during these stages of development. The inclusion of Hapln1a mutants in the manuscript as a way to support morphoHeart's capability in delivering more detailed analyses of mutant phenotypes is particularly welcomed. 

Overall, morphoHeart represents a significant step forward in the analysis of early heart morphogenesis. Its potential adaptability to other systems and organs, along with its open-access nature, make it a valuable tool for the scientific community. There are only a few minor points that require attention, and these are described below:

Figure 1: please avoid using red and green colours in the same figure (Fig 1J,) for readability and inclusivity purposes.

Figure 3: The labels around the 2D heatmaps which indicate cardiac region should be coloured in a different way (Fig 3 L-O). At the moment it is confusing to the non-specialised reader, as one could assume the green/orange/pink/blue colours could either associate with each other or even with the heatmap. Use a different set of colour pallet altogether. 

Figure 4: Please include scale bar in Fig 4J.

Figure 5: Similarly, the labels around the 2D heatmaps which indicate cardiac region should be coloured in a different way (Fig 5 K-N). At the moment it is confusing to the non-specialised reader, as one could assume the green/orange/pink/blue colours could either associate with each other or even with the heatmap. Use a different set of colour pallet altogether. For Fig 5E, please avoid using red and green colours in the same figure for readability and inclusivity purposes.

Figure 6: Similarly, the labels around the 2D heatmaps which indicate cardiac region should be coloured in a different way (Fig 6 L-O). At the moment it is confusing to the non-specialised reader, as one could assume the green/orange/pink/blue colours could either associate with each other or even with the heatmap. Use a different set of colour pallet altogether. 

Figure 7: It is not clear how many WT and hapln1a mutant hearts have been imaged and analysed in this figure to be included in the analysis presented in panels C-M. Can the authors please include this information in the figure legend, as they have done for the other figure legends?

---

## [Decision Letter · Decision Letter 2]

29 Nov 2024

Dear Dr Noël,

Thank you for your patience while we considered your revised manuscript "morphoHeart: a novel quantitative tool to perform integrated 3D morphometric analyses of heart and ECM morphology during embryonic development" for publication as a Methods and Resources Article at PLOS Biology. This revised version of your manuscript has been evaluated by the PLOS Biology editors, the Academic Editor and the original reviewers.

Based on the reviews, I am pleased to say that we are likely to accept this manuscript for publication, provided you satisfactorily address the following data and other policy-related requests that I have provided below (A-E):

(A) We would like to suggest the following minor modification to the title:

"morphoHeart: a quantitative tool for integrated 3D morphometric analyses of heart and ECM during embryonic development"

(B) You may be aware of the PLOS Data Policy, which requires that all data be made available without restriction: http://journals.plos.org/plosbiology/s/data-availability. For more information, please also see this editorial: http://dx.doi.org/10.1371/journal.pbio.1001797

-Supplementary files (e.g., excel). Please ensure that all data files are uploaded as 'Supporting Information' and are invariably referred to (in the manuscript, figure legends, and the Description field when uploading your files) using the following format verbatim: S1 Data, S2 Data, etc. Multiple panels of a single or even several figures can be included as multiple sheets in one excel file that is saved using exactly the following convention: S1_Data.xlsx (using an underscore).

-Deposition in a publicly available repository. Please also provide the accession code or a reviewer link so that we may view your data before publication. 

Figure 2J-L, 2N-Q, 3B-C, 3K-N, 4B-E, 4G-H, 5A, 5C-D, 5K-N, 6C-D, 6L-O, 6Q-R, 7C-G, 7H-N, S2G-I, S2L-N, S3C-H, S5E-G, S5J-L, S6E-G, S8H-J, S9B-C

(C) Please also ensure that each of the relevant figure legends in your manuscript include information on *WHERE THE UNDERLYING DATA CAN BE FOUND*, and ensure your supplemental data file/s has a legend.

(D) Please note that we cannot accept sole deposition of code in GitHub, as this could be changed after publication. However, you can archive this version of your publicly available GitHub code to Zenodo. Once you do this, it will generate a DOI number, which you will need to provide in the Data Accessibility Statement (you are welcome to also provide the GitHub access information). See the process for doing this here: https://docs.github.com/en/repositories/archiving-a-github-repository/referencing-and-citing-content

(E) Please ensure that your Data Statement in the submission system accurately describes where your data can be found and is in final format, as it will be published as written there. 

We expect to receive your revised manuscript within two weeks. 

*Published Peer Review History*

*Press*

Sincerely,

Richard

Richard Hodge, PhD

rhodge@plos.org

Reviewer remarks:

Reviewer #1: This is a highly responsive and very thorough revision. The authors were thoughtful in addressing the reasoning behind the requests/suggestions. They added several new panels showing different comparisons that were requested (live vs. fixed, WT vs hapln1a, negative space vs. an ECM marker), and for every critique they added written content to describe results, contextualize findings, make recommendations to potential users, etc. This excellent revision will be highly valued by the readers of PLoS Biology. 

Reviewer #2: The manuscript by Possada et al. has been considerably improved from the previous version. The authors have taken care to address comments/concerns from all reviewers thoroughly, and have added quite a few experiments that give needed insight into the functionality of morphoHeart. The section of "Practical considerations for analysis of transgenic lines" is particularly useful, as it clearly highlights the limitations of using only 1 transgenic line for phenotypic analysis and the importance of being consistent in the lines used to compare wild-type and mutant data. I have no further concerns for the manuscript. 

Reviewer #3 (Filipa Simões, signs review): The authors have done a great job addressing my comments and suggestions for improving the accessibility of their work to the community. I have no further suggestions.

---

## [Editor Report · Decision Letter 3]

20 Dec 2024

Dear Dr Noël,

On behalf of my colleagues and the Academic Editor, Simon Hughes, I am pleased to say that we can accept your manuscript for publication, provided you address any remaining formatting and reporting issues. These will be detailed in an email you should receive within 2-3 business days from our colleagues in the journal operations team; no action is required from you until then. Please note that we will not be able to formally accept your manuscript and schedule it for publication until you have completed any requested changes.

PRESS

Best wishes, 

Richard

Richard Hodge, PhD

rhodge@plos.org

PLOS
